# Evolving interpretable plasticity for spiking networks

**Jakob Jordan[1][†]\*, Maximilian Schmidt[2,3][†], Walter Senn[1], Mihai A Petrovici[1,4]**

[1]Department of Physiology, University of Bern, Bern, Switzerland; [2]Ascent Robotics, Tokyo, Japan; [3]RIKEN Center for Brain Science, Tokyo, Japan; [4]Kirchhoff-Institute for Physics, Heidelberg University, Heidelberg, Germany

**Abstract** Continuous adaptation allows survival in an ever-changing world. Adjustments in the synaptic coupling strength between neurons are essential for this capability, setting us apart from simpler, hard-wired organisms. How these changes can be mathematically described at the phenomenological level, as so-called 'plasticity rules', is essential both for understanding biological information processing and for developing cognitively performant artificial systems. We suggest an automated approach for discovering biophysically plausible plasticity rules based on the definition of task families, associated performance measures and biophysical constraints. By evolving compact symbolic expressions, we ensure the discovered plasticity rules are amenable to intuitive understanding, fundamental for successful communication and human-guided generalization. We successfully apply our approach to typical learning scenarios and discover previously unknown mechanisms for learning efficiently from rewards, recover efficient gradient-descent methods for learning from target signals, and uncover various functionally equivalent STDP-like rules with tuned homeostatic mechanisms.

**\*For correspondence:**
jakob.jordan@unibe.ch

[†]These authors contributed equally to this work

**Competing interests:** The authors declare that no competing interests exist.

## Introduction

How do we learn? Whether we are memorizing the way to the lecture hall at a conference or mastering a new sport, somehow our central nervous system is able to retain the relevant information over extended periods of time, sometimes with ease, other times only after intense practice. This acquisition of new memories and skills manifests at various levels of the system, with changes of the interaction strength between neurons being a key ingredient. Uncovering the mechanisms behind this synaptic plasticity is a key challenge in understanding brain function. Most studies approach this monumental task by searching for phenomenological models described by symbolic expressions that map local biophysical quantities to changes of the connection strength between cells (*Figure 1A,B*).

Approaches to deciphering synaptic plasticity can be broadly categorized into bottom-up and top-down. Bottom-up approaches typically rely on experimental data (e.g., *Artola et al., 1990*; *Dudek and Bear, 1993*; *Bi and Poo, 1998*; *Ngezahayo et al., 2000*) to derive dynamic equations for synaptic parameters that lead to functional emergent macroscopic behavior if appropriately embedded in networks (e.g., *Gütig et al., 2003*; *Izhikevich, 2007*; *Clopath et al., 2010*). Top-down approaches proceed in the opposite direction: from a high-level description of network function, for example, in terms of an objective function (e.g., *Toyoizumi et al., 2005*; *Deneve, 2008*; *Kappel et al., 2015*; *Kutschireiter et al., 2017*; *Sacramento et al., 2018*; *Göltz et al., 2019*), dynamic equations for synaptic changes are derived and biophysically plausible implementations suggested. Evidently, this demarcation is not strict, as most approaches seek some balance between experimental evidence, functional considerations and model complexity. However, the relative weighting of each of these aspects is usually not made explicit in the communication of scientific results, making it difficult to track by other researchers. Furthermore, the selection of specific tasks

**eLife digest** Our brains are incredibly adaptive. Every day we form memories, acquire new knowledge or refine existing skills. This stands in contrast to our current computers, which typically can only perform pre-programmed actions. Our own ability to adapt is the result of a process called synaptic plasticity, in which the strength of the connections between neurons can change. To better understand brain function and build adaptive machines, researchers in neuroscience and artificial intelligence (AI) are modeling the underlying mechanisms.

So far, most work towards this goal was guided by human intuition – that is, by the strategies scientists think are most likely to succeed. Despite the tremendous progress, this approach has two drawbacks. First, human time is limited and expensive. And second, researchers have a natural – and reasonable – tendency to incrementally improve upon existing models, rather than starting from scratch.

Jordan, Schmidt et al. have now developed a new approach based on 'evolutionary algorithms'. These computer programs search for solutions to problems by mimicking the process of biological evolution, such as the concept of survival of the fittest. The approach exploits the increasing availability of cheap but powerful computers. Compared to its predecessors (or indeed human brains), it also uses search strategies that are less biased by previous models.

The evolutionary algorithms were presented with three typical learning scenarios. In the first, the computer had to spot a repeating pattern in a continuous stream of input without receiving feedback on how well it was doing. In the second scenario, the computer received virtual rewards whenever it behaved in the desired manner – an example of reinforcement learning. Finally, in the third 'supervised learning' scenario, the computer was told exactly how much its behavior deviated from the desired behavior. For each of these scenarios, the evolutionary algorithms were able to discover mechanisms of synaptic plasticity to solve the new task successfully.

Using evolutionary algorithms to study how computers 'learn' will provide new insights into how brains function in health and disease. It could also pave the way for developing intelligent machines that can better adapt to the needs of their users.

to illustrate the effect of a suggested learning rule is usually made only after the rule was derived based on other considerations. Hence, this typically does not consider competing alternative solutions, as an exhaustive comparison would require significant additional investment of human resources. A related problem is that researchers, in a reasonable effort to use resources efficiently, tend to focus on promising parts of the search space around known solutions, leaving large parts of the search space unexplored (*Radi and Poli, 2003*). Automated procedures, in contrast, can perform a significantly less biased search.

We suggest an automated approach to discover learning rules in spiking neuronal networks that explicitly addresses these issues. Automated procedures interpret the search for biological plasticity mechanisms as an optimization problem (*Bengio et al., 1992*), an idea typically referred to as meta-learning or learning-to-learn. These approaches make the emphasis of particular aspects that guide this search explicit and place the researcher at the very end of the process, supporting much larger search spaces and the generation of a diverse set of hypotheses. Furthermore, they have the potential to discover domain-specific solutions that are more efficient than general-purpose algorithms. Early experiments focusing on learning in artificial neural networks (ANNs) made use of gradient descent or genetic algorithms to optimize parameterized learning rules (*Bengio et al., 1990*; *Bengio et al., 1992*; *Bengio et al., 1993*) or genetic programming to evolve less constrained learning rules (*Bengio et al., 1994*; *Radi and Poli, 2003*), rediscovering mechanisms resembling the backpropagation of errors (*Linnainmaa, 1970*; *Ivakhnenko, 1971*; *Rumelhart et al., 1985*). Recent experiments demonstrate how optimization methods can design optimization algorithms for recurrent ANNs (*Andrychowicz et al., 2016*), evolve machine learning algorithms from scratch (*Real et al., 2020*), and optimize parametrized learning rules in neuronal networks to achieve a desired function (*Confavreux et al., 2020*).

We extend these meta-learning ideas to discover free-form, yet interpretable plasticity rules for spiking neuronal networks. The discrete nature of spike-based neuronal interactions endows these

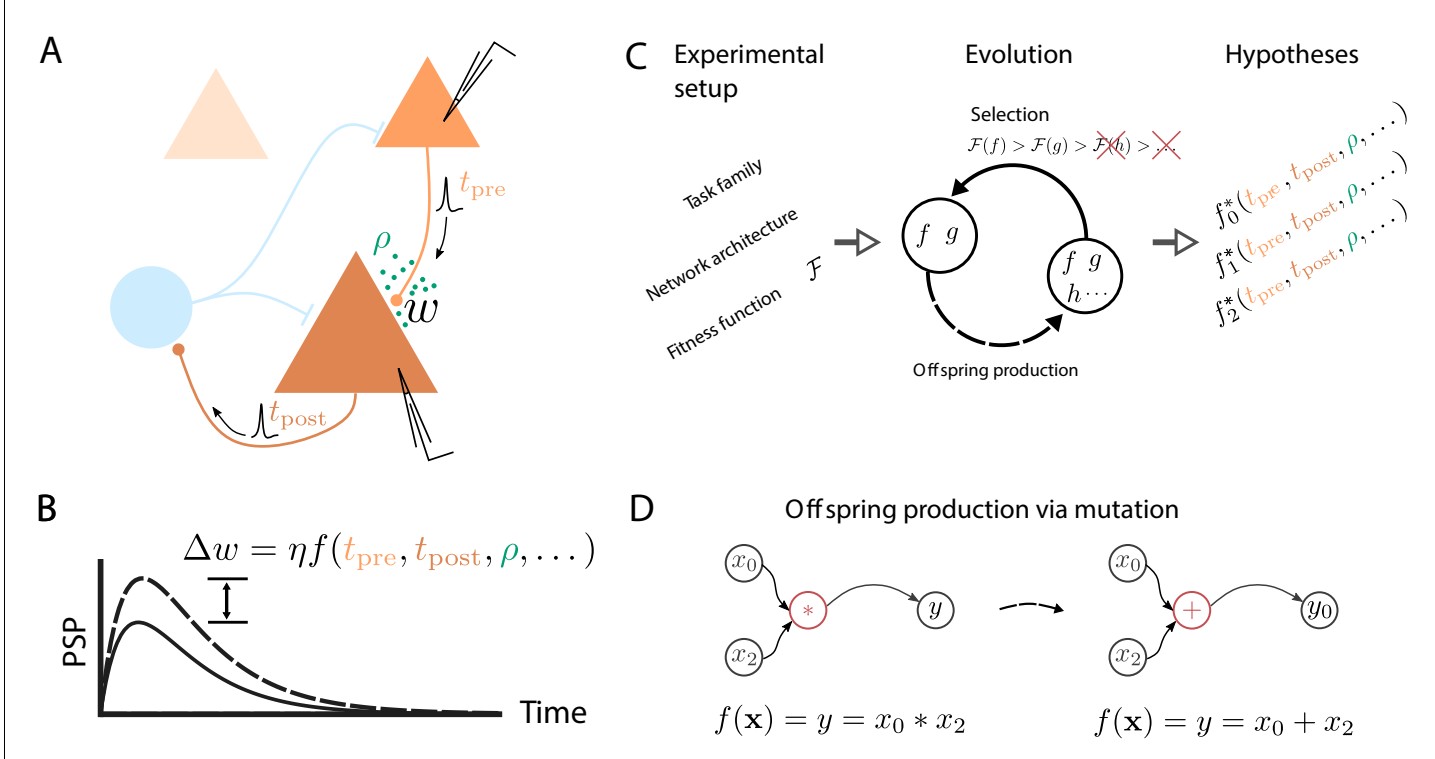

**Figure 1.** Artificial evolution of synaptic plasticity rules in spiking neuronal networks. (A) Sketch of cortical microcircuits consisting of pyramidal cells (orange) and inhibitory interneurons (blue). Stimulation elicits action potentials in pre- and postsynaptic cells, which, in turn, influence synaptic plasticity. (B) Synaptic plasticity leads to a weight change ($\Delta w$) between the two cells, here measured by the change in the amplitude of post-synaptic potentials. The change in synaptic weight can be expressed by a function $f$ that in addition to spike timings ($t_{pre}, t_{post}$) can take into account additional local quantities, such as the concentration of neuromodulators ($\rho$, green dots in A) or postsynaptic membrane potentials. (C) For a specific experimental setup, an evolutionary algorithm searches for individuals representing functions $f$ that maximize the corresponding fitness function $\mathcal{F}$. An offspring is generated by modifying the genome of a parent individual. Several runs of the evolutionary algorithm can discover phenomenologically different solutions ($f_0, f_1, f_2$) with comparable fitness. (D) An offspring is generated from a single parent via mutation. Mutations of the genome can, for example, exchange mathematical operators, resulting in a different function $f$.

networks with rich dynamical and functional properties (e.g., *Dold et al., 2019*; *Jordan et al., 2019*; *Keup et al., 2020*). In addition, with the advent of non-von Neumann computing systems based on spiking neuronal networks with online learning capabilities (*Moradi et al., 2017*; *Davies et al., 2018*; *Billaudelle et al., 2019*), efficient learning algorithms for spiking systems become increasingly relevant for non-conventional computing. Here, we employ genetic programming (*Figure 1C,D*; *Koza, 2010*) as a search algorithm for two main reasons. First, genetic programming can operate on analytically tractable mathematical expressions describing synaptic weight changes that are interpretable. Second, an evolutionary search does not need to compute gradients in the search space, thereby circumventing the need to estimate a gradient in non-differentiable systems.

We successfully apply our approach, which we refer to as 'evolving-to-learn' (E2L), to three different learning paradigms for spiking neuronal networks: reward-driven, error-driven, and correlation-driven learning. For the reward-driven task, our approach discovers new plasticity rules with efficient reward baselines that perform competively and even outperform previously suggested methods. The analytic form of the resulting expressions suggests experimental approaches that would allow us to distinguish between them. In the error-driven learning scenario, the evolutionary search discovers a variety of solutions which, with appropriate analysis of the corresponding expressions, can be shown to effectively implement stochastic gradient descent. Finally, in the correlation-driven task, our method generates a variety of STDP kernels and associated homeostatic mechanisms that lead to similar network-level behavior. This sheds new light onto the observed variability of synaptic

plasticity and thus suggests a reevaluation of the reported variety in experimentally measured STDP curves with respect to their possible functional equivalence.

Our results demonstrate the significant potential of automated procedures in the search for plasticity rules in spiking neuronal networks, analogous to the transition from hand-designed to learned features that lies at the heart of modern machine learning.

## Results

### Setting up an evolutionary search for plasticity rules

We introduce the following recipe to search for biophysically plausible plasticity rules in spiking neuronal networks. First, we determine a task family of interest and an associated experimental setup which includes specification of the network architecture, for example, neuron types and connectivity, as well as stimulation protocols or training data sets. Crucially, this step involves defining a fitness function to guide the evolutionary search towards promising regions of the search space. It assigns high fitness to those individuals, that is, learning rules, that solve the task well and low fitness to others. The fitness function may additionally contain constraints implied by experimental data or arising from computational considerations. We determine each individual's fitness on various examples from the given task family, for example, different input spike train realizations, to discover plasticity rules that generalize well (*Chalmers, 1991*; *Soltoggio et al., 2018*). Finally, we specify the neuronal variables available to the plasticity rule, such as low-pass-filtered traces of pre- and postsynaptic spiking activity or neuromodulator concentrations. This choice is guided by biophysical considerations, for example, which quantities are locally available at a synapse, as well as by the task family, for example, whether reward or error signals are provided by the environment. We write the plasticity rule in the general form $\Delta w = \eta f(\ldots)$, where $\eta$ is a fixed learning rate, and employ an evolutionary search to discover functions $f$ that lead to high fitness.

We propose to use genetic programming (GP) as an evolutionary algorithm to discover plasticity rules in spiking neuronal networks. GP applies mutations and selection pressure to an initially random population of computer programs to artificially evolve algorithms with desired behaviors (e.g., *Koza, 1992*). Here, we consider the evolution of mathematical expressions. We employ a specific form of GP, Cartesian genetic programming (CGP; e.g., *Miller and Thomson, 2000*; *Miller, 2011*), that uses an indexed graph representation of programs. The genotype of an individual is a two-dimensional Cartesian graph (*Figure 2A*, top). Over the course of an evolutionary run, this graph has a fixed number of input, output, and internal nodes. The operation of each internal node is fully described by a single function gene and a fixed number of input genes. A function table maps function genes to mathematical operations (*Figure 2A*, bottom), while input genes determine from where this node receives data. A given genotype is decoded into a corresponding computational graph (the phenotype, *Figure 2B*) which defines a function $f$. During the evolutionary run, mutations of the genotype alter connectivity and node operations, which can modify the encoded function (*Figure 2C*). The indirect encoding of the computational graph via the genotype supports variable-length phenotypes, since some internal nodes may not be used to compute the output (*Figure 2B*). The size of the genotype, in contrast, is fixed, thereby restricting the maximal size of the computational graph and ensuring compact, interpretable mathematical expressions. Furthermore, the separation into genotype and phenotype allows the buildup of 'silent mutations', that is, mutations in the genotype that do not alter the phenotype. These silent mutations can lead to more efficient search as they can accumulate and in combination lead to an increase in fitness once affecting the phenotype (*Miller and Thomson, 2000*). A $\mu + \lambda$ evolution strategy (*Beyer and Schwefel, 2002*) drives evolution by creating the next generation of individuals from the current one via tournament selection, mutation and selection of the fittest individuals (see section Evolutionary algorithm). Prior to starting the search, we choose the mathematical operations that can appear in the plasticity rule and other (hyper)parameters of the Cartesian graph and evolutionary algorithm. For simplicity, we consider a restricted set of mathematical operations and additionally make use of nodes with constant output. After the search has completed, for example, by reaching a target fitness value or a maximal number of generations, we analyze the discovered set of solutions.

In the following, we describe the results of three experiments following the recipe outlined above.

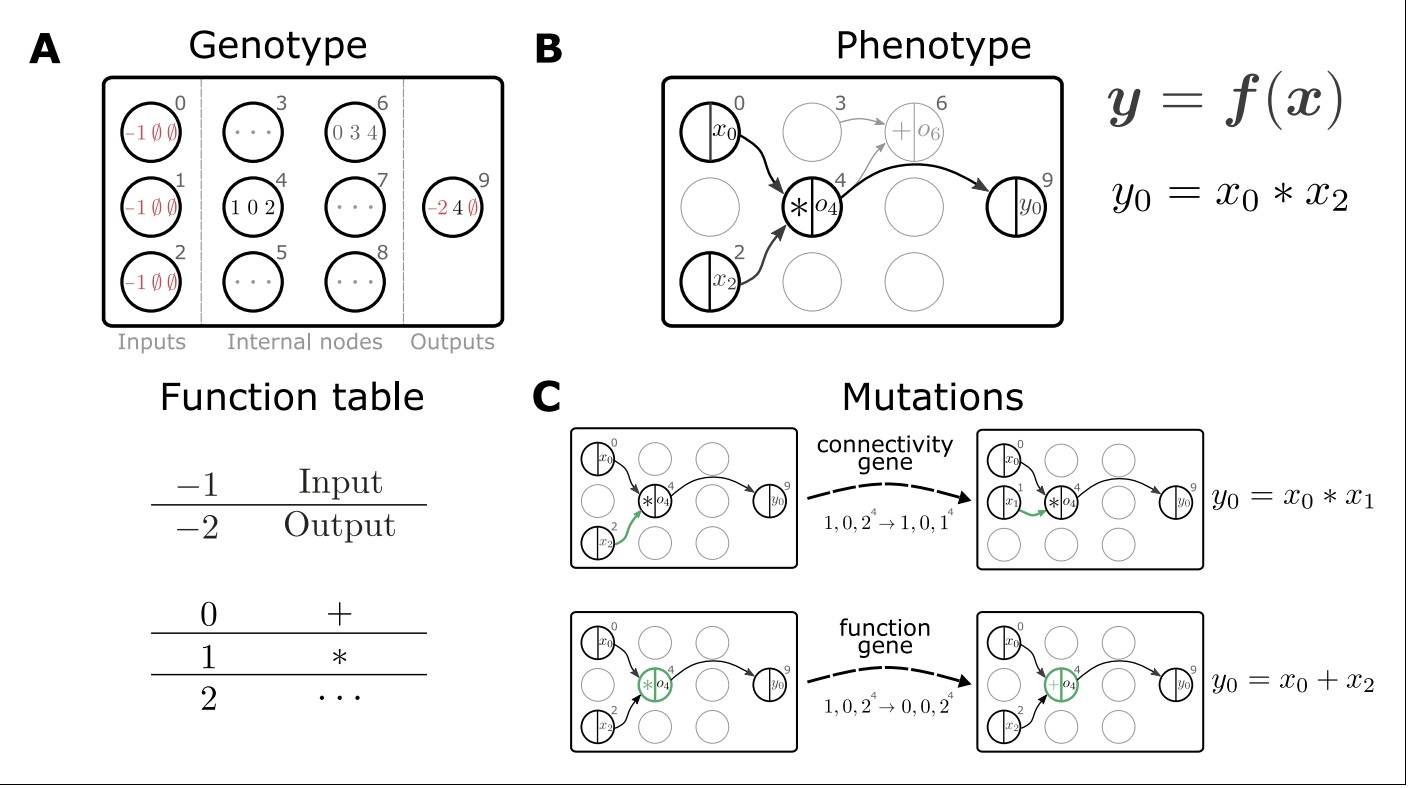

**Figure 2.** Representation and mutation of mathematical expressions in Cartesian genetic programming. (A) The genotype of an individual is a two-dimensional Cartesian graph (top). In this example, the graph contains three input nodes ($0-2$), six internal nodes ($3-8$) and a single output node ($9$). In each node, the genes of a specific genotype are shown, encoding the operator used to compute the node's output and its inputs. Each operator gene maps to a specific mathematical function (bottom). Special values ($-1, -2$) represent input and output nodes. For example, node four uses the operator 1, the multiplication operation '*', and receives input from nodes 0 and 2. This node's output is hence given by $x_0 * x_2$. The number of input genes per node is determined by the operator with the maximal arity (here two). Fixed genes that cannot be mutated are highlighted in red. $\emptyset$ denotes non-coding genes. (B) The computational graph (phenotype) generated by the genotype in A. Input nodes ($x_0, x_1, x_2$) represent the arguments of the function $f$. Each output node selects one of the other nodes as a return value of the computational graph, thus defining a function from input $\boldsymbol{x}$ to output $\boldsymbol{y} = \boldsymbol{f}(\boldsymbol{x})$. Here, the output node selects node four as a return value. Some nodes defined in the genotype are not used by a particular realization of the computational graph (in light gray, e.g., node 6). Mutations that affect such nodes have no effect on the phenotype and are therefore considered 'silent'. (C) Mutations in the genome either lead to a change in graph connectivity (top, green arrow) or alter the operators used by an internal node (bottom, green node). Here, both mutations affect the phenotype and are hence not silent.

## Evolving an efficient reward-driven plasticity rule

We consider a simple reinforcement learning task for spiking neurons. The experiment can be succinctly described as follows: $N$ inputs project to a single readout modeled by a leaky integrator neuron with exponential postsynaptic currents and stochastic spike generation (for details see section Reward-driven learning task). We generate a finite number $M$ of frozen-Poisson-noise patterns of duration $T$ and assign each of these randomly to one of two classes. The output neuron should learn to classify each of these spatio-temporal input patterns into the corresponding class using a spike/no-spike code (**Figure 3A,B**).

The fitness $\mathcal{F}(f)$ of an individual encoding the function $f$ is measured by the mean reward per trial averaged over a certain number of experiments $n_{\mathrm{exp}}$, each consisting of $n$ classification trials

$$\mathcal{F}(f) := \frac{1}{n_{\mathrm{exp}}} \sum_{k=1}^{n_{\mathrm{exp}}} R_k(f)\,, \tag{1}$$

where $R_k(f) := \frac{1}{n}\sum_{i=1}^{n} R_{k,i}(f)$ is the mean reward per trial obtained in experiment $k$. The reward $R_{k,i}$ is the reward obtained in the $i$ th trial of experiment $k$. It is one if the classification is correct and -1 otherwise. In the following, we drop the subscripts from $R_{k,i}$ where its meaning is clear from context.

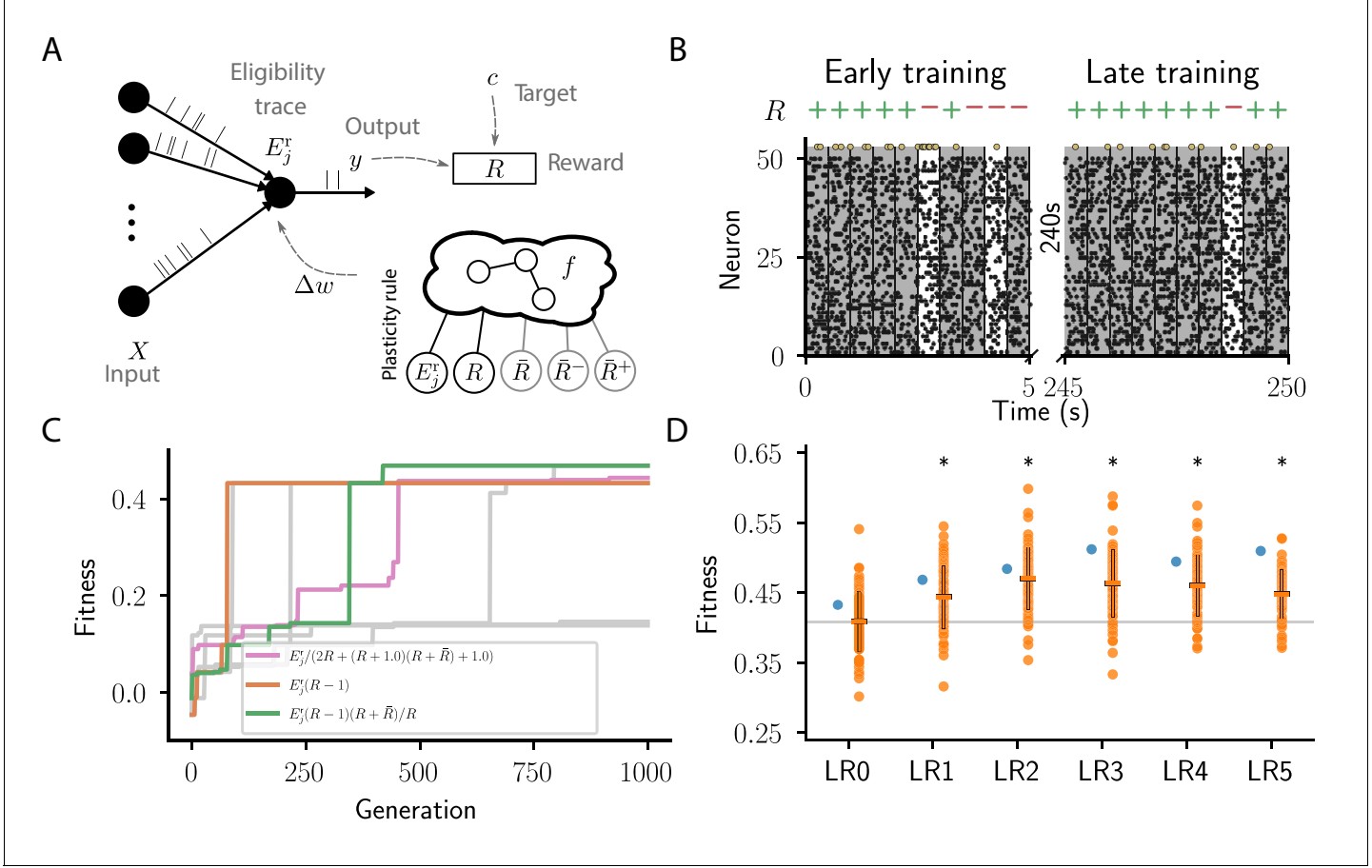

**Figure 3.** Cartesian genetic programming evolves various efficient reward-driven learning rules. (A) Network sketch. Multiple input neurons with Poisson activity project to a single output unit. Pre- and postsynaptic activity generate an eligibility trace in each synapse. Comparison between the output activity and the target activity generates a reward signal. $\bar{R}$, and $\bar{R}^+$, $\bar{R}^-$ represent the expected reward, the expected positive and the expected negative reward, respectively. Depending on the hyperparameter settings either the former or the latter two are provided to the plasticity rule. (B) Raster plot of the activity of input neurons (small black dots) and output neuron (large golden dots). Gray (white) background indicate patterns for which the output should be active (inactive). Top indicates correct classifications (+) and incorrect classifications (-). We show 10 trials at the beginning (left) and the end of training (right) using the evolved plasticity rule: $\Delta w_j = \eta\,(R-1)E_j^r$. (C) Fitness of best individual per generation as a function of the generation index for multiple example runs of the evolutionary algorithm with different initial conditions but identical hyperparameters. Labels show the expression $f$ at the end of the respective run for three runs resulting in well-performing plasticity rules. Gray lines represent runs with functionally identical solutions or low final fitness. (D) Fitness of a selected subset of evolved learning rules on the 10 experiments used during the evolutionary search (blue) and additional 80 fitness evaluations, each on 10 new experiments consisting of sets of frozen noise patterns and associated class labels not used during the evolutionary search (orange). Horizontal boxes represent mean, error bars indicate one standard deviation over fitness values. Gray line indicates mean fitness of LR0 for visual reference. Black stars indicate significance ($p<10^{-16}$) with respect to LR0 according to Welch's T-tests (**Welch, 1947**). See main text for the full expressions for all learning rules.

Each experiment contains different realizations of frozen-noise input spike trains with associated class labels.

Previous work on reward-driven learning (**Williams, 1992**) has demonstrated that in policy-gradient-based approaches (e.g., **Sutton and Barto, 2018**), subtracting a so called 'reward baseline' from the received reward can improve the convergence properties by adjusting the magnitude of weight updates. However, choosing a good reward baseline is notoriously difficult (**Williams, 1988**; **Dayan, 1991**; **Weaver and Tao, 2001**). For example, in a model for reinforcement learning in spiking neurons, **Vasilaki et al., 2009** suggest the expected positive reward as a suitable baseline. Here, we consider plasticity rules which, besides immediate rewards, have access to expected rewards. These expectations are obtained as moving averages over a number of consecutive trials (here: 100

trials, i.e., 50 s) during one experiment and can either be estimated jointly ($\bar{R} \in [-1, 1]$) or separately for positive ($\bar{R}^+ \in [0, 1]$) and negative ($\bar{R}^- \in [-1, 0]$) rewards, with $\bar{R} = \bar{R}^+ + \bar{R}^-$ (for details, see section Reward-driven learning task). In the former case, the plasticity rule takes the general form

$$\Delta w_j = \eta f\left(R, E_j^{\mathrm{r}}(T), \bar{R}\right), \tag{2}$$

while for seperately estimated positive and negative rewards it takes the form

$$\Delta w_j = \eta f\left(R, E_j^{\mathrm{r}}(T), \bar{R}^+, \bar{R}^-\right). \tag{3}$$

In both cases, $\eta$ is a fixed learning rate and $E_j^{\mathrm{r}}(t)$ is an eligibility trace that contains contributions from the spiking activity of pre- and post-synaptic neurons which is updated over the course of a single trial (for details see section Reward-driven learning task). The eligibility trace arises as a natural consequence of policy-gradient methods aiming to maximize the expected reward (*Williams, 1992*) and is a common ingredient of reward-modulated plasticity rules for spiking neurons (*Vasilaki et al., 2009*; *Frémaux and Gerstner, 2015*). It is a low-pass filter of the product of two terms: the first is positive if the neuron was more active than expected by synaptic input; this can happen because the neuronal output is stochastic, to promote exploration. The second is a low-pass filter of presynaptic activity. A simple plasticity rule derived from maximizing the expected rewards would, for example, change weights according to the product of the received reward and the eligibility trace: $\Delta w_j = R E_j^{\mathrm{r}}$. If by chance a neuron is more active than expected, and the agent receives a reward, all weights of active afferents are increased, making it more likely for the neuron to fire in the future given identical input. Reward and eligibility trace values at the end of each trial ($t = T$) are used to determine synaptic weight changes. In the following, we drop the time argument of $E_j^{\mathrm{r}}$ for simplicity. Using CGP with three ($R, E_j^{\mathrm{r}}, \bar{R}$), or four inputs ($R, E_j^{\mathrm{r}}, \bar{R}^+, \bar{R}^-$), respectively, we search for plasticity rules that maximize the fitness $\mathcal{F}(f)$ (*Equation 1*).

None of the evolutionary runs with access to the expected reward ($\bar{R}$) make use of it as a reward baseline (see Appendix section Full evolution data for different CGP hyperparameter choices for full data). Some of them discover high-performing rules identical to that suggested by *Urbanczik and Senn, 2009*: $\Delta w_j = \eta (R - 1) E_j^{\mathrm{r}}$ (LR0, $\mathcal{F} = 216.2$, *Figure 3C,D*). This rule uses a fixed base line, the maximal reward ($R_{\max} = 1$), rather than the expected reward. Some runs discover a more sophisticated variant of this rule with a term that decreases the effective learning rate for negative rewards as the agent improves, that is, when the expected reward increases: $\Delta w_j = \eta (1 + R\bar{R})(R - 1) E_j^{\mathrm{r}}$ (LR1, $\mathcal{F} = 234.2$, *Figure 3C,D*; see also Appendix section Causal and homeostatic terms over trials). Using this effective learning-rate, this rule achieve higher fitness than the vanilla formulation at the expense of requiring the agent to keep track of the expected reward.

Using the expected reward as a baseline, for example, $\Delta w_j = \eta (R - \bar{R}) E_j^{\mathrm{r}}$, is unlikely to yield high-performing solutions: an agent may get stuck in weight configurations in which it continuously receives negative rewards, yet, as it is expecting negative rewards, does not significantly change its weights. This intuition is supported by our observation that none of the high-performing plasticity rules discovered by our evolutionary search make use of such a baseline, in contrast to previous studies (e.g., *Frémaux and Gerstner, 2015*). Subtracting the maximal reward, in contrast, can be interpreted as an optimistic baseline (cf. *Sutton and Barto, 2018*, Ch2.5), which biases learning to move away from weight configurations that result in negative rewards, while maintaining weight configurations that lead to positive rewards. However, a detrimental effect of such an optimistic baseline is that learning is sparse, as it only occurs upon receiving negative rewards, an assumption at odds with behavioral evidence.

In contrast, evolutionary runs with access to separate estimates of the negative and positive rewards discover plasticity rules with efficient baselines, resulting in increased fitness (see Appendix section Full evolution data for different CGP hyperparameter choices for the full data). In the following, we discuss four such high-performing plasticity rules with at least 10% higher fitness than LR0 (*Figure 3D*). We first consider the rule (LR2, $\mathcal{F} = 242.0$, *Figure 3D*)

$$\Delta w_j = \eta [R - (\bar{R}^+ - \bar{R}^-)] E_j^{\mathrm{r}} = \eta (R - \bar{R}_{\mathrm{abs}}) E_j^{\mathrm{r}}, \tag{4}$$

where we introduced the expected absolute reward $\bar{R}_{\mathrm{abs}} := \bar{R}^+ - \bar{R}^- = |\bar{R}^+| + |\bar{R}^-|$, with $\bar{R}_{\mathrm{abs}} \in [0, 1]$. Note the difference to the expected reward $\bar{R} = \bar{R}^+ + \bar{R}^-$. Since the absolute magnitude of positive and negative rewards is identical in the considered task, $\bar{R}_{\mathrm{abs}}$ increases in each trial, starting at zero and slowly converging to one with a time constant of 50 s. Instead of keeping track of the expected reward, the agent can thus simply optimistically increase its baseline with each trial. Behind this lies the, equally optimistic, expectation that the agent improves its performance over trials. Starting out as $RE_j^{\mathrm{r}}$ and converging to $(R-1)E_j^{\mathrm{r}}$ this rule combines efficient learning from both positive and negative rewards initially, with improved convergence towards successful weight configuration during late learning by a reward-dependent modulation of the effective learning rate (see also Appendix section Causal and homeostatic terms over trials). Note that such a strategy may lead to issues with un- or re-learning. This may be overcome by the agent resetting the expected absolute reward $\bar{R}_{\mathrm{abs}}$ upon encountering a new task, similar to a 'novelty' signal.

Furthermore, our algorithm discovers a variation of this rule (LR3, $\mathcal{F} = 256.0$, **Figure 3D**), which replaces $\eta$ with $\eta/(1 + \bar{R}^+)$ to decrease the magnitude of weight changes in regions of the weight space associated with high performance. This can improve convergence properties.

We next consider the rule (LR4, $\mathcal{F} = 247.2$, **Figure 3D**):

$$\Delta w_j = \eta \left[ (R-1)E_j^{\mathrm{r}} + (R-1)(R + 2\bar{R}^+) \right]. \tag{5}$$

This rule has the familiar form of LR0 and LR1, with an additional homeostatic term. Due to the prefactors $R-1$, this rule only changes weights on trials with negative reward. Initially, the expected reward $\bar{R}^+$ is close to zero and the homeostatic term results in potentiation of all synapses, causing more and more neurons to spike. In particular, if initial weights are chosen poorly, this can make learning more robust. As the agent improves and the expected positive rewards increases, the homeostatic term becomes negative (see also Appendix section Causal and homeostatic terms over trials). In this regime, it can be interpreted as pruning all weights until only those are left that do not lead to negative rewards. This term can hence be interpreted as an adapting action baseline (**Sutton and Barto, 2018**).

Finally, we consider the rule (LR5, $\mathcal{F} = 254.8$, **Figure 3D**):

$$\Delta w_j = \eta \left\{ 2[R - (\bar{R}^+ - R\bar{R}^-)]E_j^{\mathrm{r}} - [R - (\bar{R}^+ - R\bar{R}^-)]R\bar{R}^- \right\}. \tag{6}$$

To analyze this seemingly complex rule, we consider the expression for trials with positive and trials with negative reward separately:

$$R = 1: \ \Delta w_j^+ = \eta \left\{ 2(1 - \bar{R}_{\mathrm{abs}})E_j^{\mathrm{r}} - (1 - \bar{R}_{\mathrm{abs}})\bar{R}^- \right\},$$
$$R = -1: \ \Delta w_j^- = \eta \left\{ 2(-1 - \bar{R})E_j^{\mathrm{r}} - (1 + \bar{R})\bar{R}^- \right\}.$$

Both expressions contain a 'causal' term depending on pre- and postsynaptic activity via the eligibility trace, as well as, and a 'homeostatic' term. Aside from the constant scaling factor, the causal term of $\Delta w_j^+$ is identical to LR2 (**Equation 4**), that is, it only causes weight changes early during learning, and converges to zero for later times. Similarly, the causal term of $\Delta w_j^-$ is initially identical to that of LR2 (**Equation 4**), decreasing weights for connections contributing to wrong decisions. However it increases in magnitude as the agent improves and the expected reward increases. The homeostatic term of $\Delta w_j^+$ is potentiating, similarly to LR4 (**Equation 5**): it encourages spiking by increasing all synaptic weights during early learning and decreases over time. The homeostatic term for negative rewards is also potentiating, but does not vanish for long times unless the agent is performing perfectly ($\bar{R}^- \to 0$). Over time, this plasticity rule hence reacts less and less to positive rewards, while increasing weight changes for negative rewards. The reward-modulated potentiating homeostatic mechanisms can prevent synaptic weights from vanishing and thus encourage exploration for challenging scenarios in which the agent mainly receives negative rewards.

In conclusion, by making use of the separately estimated expected negative and positive rewards in precise combinations with the eligibility trace and the instantaneous reward, our evolving-to-learn approach discovered a variety of reward-based plasticity rules, many of them outperforming previously known solutions (e.g., **Urbanczik and Senn, 2009**). The evolution of closed-form expressions

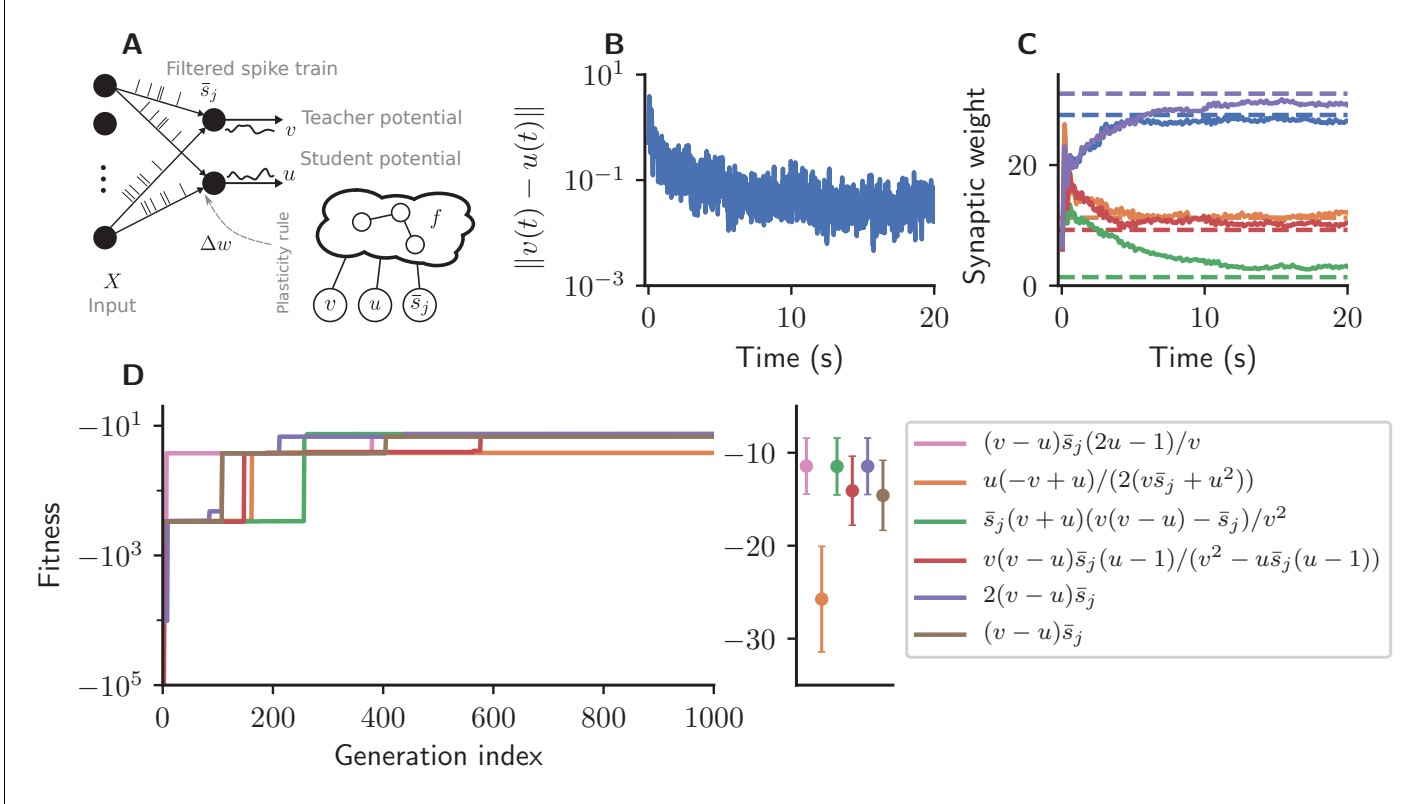

**Figure 4.** Cartesian genetic programming evolves efficient error-driven learning rules. (**A**) Network sketch. Multiple input neurons with Poisson activity project to two neurons. One of the neurons (the teacher) generates a target for the other (the student). The membrane potentials of teacher and student as well as the filtered pre-synaptic spike trains are provided to the plasticity rule that determines the weight update. (**B**) Root mean squared error between the teacher and student membrane potential over the course of learning using the evolved plasticity rule: $\Delta w_j(t) = \eta \left[ v(t) - u(t) \right] \bar{s}_j(t)$. (**C**) Synaptic weights over the course of learning corresponding to panel B. Horizontal dashed lines represent target weights, that is, the fixed synaptic weights onto the teacher. (**D**) Fitness of the best individual per generation as a function of the generation index for multiple runs of the evolutionary algorithm with different initial conditions. Labels represent the rule at the end of the respective run. Colored markers represent fitness of each plasticity rule averaged over 15 validation tasks not used during the evolutionary search; error bars indicate one standard deviation.

allowed us to analyze the computational principles that allow these newly discovered rules to achieve high fitness. This analysis suggests new mechanisms for efficient learning, for example from 'novelty' and via reward-modulated homeostatic mechanisms. Each of these new hypotheses for reward-driven plasticity rules makes specific predictions about behavioral and neuronal signatures that potentially allow us to distinguish between them. For example LR2, LR3, and LR5 suggest that agents initially learn both from positive and negative rewards, while later they mainly learn from negative rewards. In particular the initial learning from positive rewards distinguishes these hypotheses from LR0, LR1, and LR4, and previous work (**Urbanczik and Senn, 2009**). As LR2 does not make use of the, separately estimated, expected rewards, it is potentially employed in settings in which such estimates are difficult to obtain. Furthermore, LR4 and LR5 suggest that precisely regulated homeostatic mechanisms play a crucial role besides weight changes due to pre- and post-synaptic activity traces. During early learning, both rules implement potentiating homeostatic mechanisms triggered by negative rewards, likely to explore many possible weight configurations which may support successful behavior. During late learning, LR4 suggests that homeostatic changes become depressing, thus pruning unnecessary or even harmful connections. In contrast, they remain positive for LR5, potentially avoiding catastrophic dissociation between inputs and outputs for challenging tasks. Besides experimental data from the behavioral and neuronal level, different artificial reward-learning

scenarios could further further select for strengths or against weaknesses of the discovered rules. Furthermore, additional considerations, for example achieving small variance in weight updates (*Williams, 1986*; *Dayan, 1991*), may lead to particular rules being favored over others. We thus believe that our new insights into reinforcement learning are merely a forerunner of future experimental and theoretical work enabled by our approach.

## Evolving an efficient error-driven plasticity rule

We next consider a supervised learning task in which a neuron receives information about how far its output is from the desired behavior, instead of just a scalar reward signal as in the previous task. The widespread success of this approach in machine learning highlights the efficacy of learning from errors compared to correlation- or reward-driven learning (*Goodfellow et al., 2016*). It has therefore often been hypothesized that evolution has installed similar capabilities in biological nervous systems (see, e.g. *Marblestone et al., 2016*; *Whittington and Bogacz, 2019*).

*Urbanczik and Senn, 2014* introduced an efficient, flexible, and biophysically plausible implementation of error-driven learning via multi-compartment neurons. For simplicity, we consider an equivalent formulation of this learning principle in terms of two point neurons modeled as leaky integrator neurons with exponential postsynaptic currents and stochastic spike generation. One neuron mimics the somatic compartment and provides a teaching signal to the other neuron acting as the dendritic compartment. Here, the difference between the teacher and student membrane potentials drives learning:

$$\Delta w_j(t) = \eta \left[ v(t) - u(t) \right] \bar{s}_j(t) \, , \tag{7}$$

where $v$ is the teacher potential, $u$ the student membrane potential, and $\eta$ a fixed learning rate. $\bar{s}_j(t) = (\kappa * s_j)(t)$ represents the the presynaptic spike train $s_j$ filtered by the synaptic kernel $\kappa$. *Equation 7* can be interpreted as stochastic gradient descent on an appropriate cost function (*Urbanczik and Senn, 2014*) and can be extended to support credit assignment in hierarchical neuronal networks (*Sacramento et al., 2018*). For simplicity, we assume the student has direct access to the teacher's membrane potential, but in principle one could also employ proxies such as firing rates as suggested in *Pfister et al., 2010*; *Urbanczik and Senn, 2014*.

We consider a one-dimensional regression task in which we feed random Poisson spike trains into the two neurons (*Figure 4A*).

The teacher maintains fixed input weights while the student's weights should be adapted over the course of learning such that its membrane potential follows the teacher's (*Figure 4B,C*). The fitness $\mathcal{F}(f)$ of an individual encoding the function $f$ is measured by the root mean-squared error between the teacher and student membrane potential over the simulation duration $T$, excluding the initial 10%, averaged over $n_{\exp}$ experiments:

$$\mathcal{F}(f) := \frac{1}{n_{\exp}} \sum_{k=1}^{n_{\exp}} \sqrt{\int_{0.1T}^{T} \mathrm{d}t \left[ v_k(t) - u_k(t) \right]^2} \, . \tag{8}$$

Each experiment consists of different input spike trains and different teacher weights. The general form of the plasticity rule for this error-driven learning task is given by:

$$\Delta w_j = \eta f(v, u, \bar{s}_j) \, . \tag{9}$$

Using CGP with three inputs $(v, u, \bar{s}_j)$, we search for plasticity rules that maximize the fitness $\mathcal{F}(f)$.

Starting from low fitness, about half of the evolutionary runs evolve efficient plasticity rules (*Figure 4D*) closely matching the performance of the stochastic gradient descent rule of *Urbanczik and Senn, 2014*. While two runs evolve exactly *Equation 7*, other solutions with comparable fitness are discovered, such as

$$\Delta w_j = \eta (v - u) \bar{s}_j \frac{2u - 1}{v} \, , \text{and} \tag{10}$$

$$\Delta w_j = \eta \bar{s}_j (v + u) \frac{v(v - u) - \bar{s}_j}{v^2} \, . \tag{11}$$

At first sight, these rules may appear more complex, but for the considered parameter regime under the assumptions $v \approx u; v, u \gg 1$, we can write them as (see Appendix section Error-driven learning – simplification of the discovered rules):

$$\Delta w_j = \eta\, c_1 (v - u) \bar{s}_j + c_2 \,, \tag{12}$$

with a multiplicative constant $c_1 = \mathcal{O}(1)$ and a negligible additive constant $c_2$. Elementary manipulations of the expressions found by CGP thus demonstrate the similarity of these superficially different rules to *Equation 7*. Consequently, they can be interpreted as approximations of gradient descent. The constants generally fall into two categories: fine-tuning of the learning rate for the set of task samples encountered during evolution ($c_1$), which could be responsible for the slight increase in performance, and factors that have negligible influence and would most likely be pruned over longer evolutionary timescales ($c_2$). This pruning could be accelerated, for example, by imposing a penalty on the model complexity in the fitness function, thus preferentially selecting simpler solutions.

In conclusion, the evolutionary search rediscovers variations of a human-designed efficient gradient-descent-based learning rule for the considered error-driven learning task. Due to the compact, interpretable representation of the plasticity rules we are able to analyze the set of high-performing solutions and thereby identify phenomenologically identical rules despite their superficial differences.

## Evolving an efficient correlation-driven plasticity rule

We now consider a task in which neurons do not receive any feedback from the environment about their performance but instead only have access to correlations between pre- and postsynaptic activity. Specifically, we consider a scenario in which an output neuron should discover a repeating frozen-noise pattern interrupted by random background spikes using a combination of spike-timing-dependent plasticity and homeostatic mechanisms. Our experimental setup is briefly described as follows: $N$ inputs project to a single output neuron (*Figure 5A*).

The activity of all inputs is determined by a Poisson process with a fixed rate. A frozen-noise activity pattern of duration $T_{\text{pattern}}$ ms is generated once and replayed every $T_{\text{inter}}$ ms (*Figure 5B*) while inputs are randomly spiking in between.

We define the fitness $\mathcal{F}(f)$ of an individual encoding the function $f$ by the minimal average signal-to-noise ratio (SNR) across $n_{\text{exp}}$ experiments:

$$\mathcal{F}(f) := \min_k \left\{ \text{SNR}_k, k \in [1, n_{\text{exp}}] \right\} \,. \tag{13}$$

The signal-to-noise ratio $\text{SNR}_k$, following *Masquelier, 2018*, is defined as the difference between the maximal free membrane potential during pattern presentation averaged over multiple presentations ($\langle u_{k,i,\text{max}} \rangle$) and the mean of the free membrane potential in between pattern presentations ($\langle u_{k,\text{inter}} \rangle$) divided by its variance ($\text{Var}(v_{k,\text{inter}})$):

$$\text{SNR}_k := \frac{\langle u_{k,i,\text{max}} \rangle - \langle u_{k,\text{inter}} \rangle}{\text{Std}(u_{k,\text{inter}})} \,. \tag{14}$$

The free membrane potential is obtained in a separate simulation with frozen weights by disabling the spiking mechanism for the output neuron. This removes measurement noise in the signal-to-noise ratio arising from spiking and subsequent membrane-potential reset. Each experiment consists of different realizations of a frozen-noise pattern and background spiking.

We evolve learning rules of the following general form, which includes a dependence on the current synaptic weight in line with previously suggested STDP rules (*Gütig et al., 2003*):

$$\Delta w_j^{\text{STDP}} = \eta \begin{cases} f_{\text{dep}}(w_j, E_j^c) & \Delta t_j < 0 \\ f_{\text{fac}}(w_j, E_j^c) & \Delta t_j \geq 0 \,. \end{cases} \tag{15}$$

Here, $E_j^c := e^{-|\Delta t_j|/\tau}$ represents an eligibility trace that depends on the relative timing of post- and presynaptic spiking ($\Delta t_j = t_{\text{post}} - t_{\text{pre},j}$) and is represented locally in each synapse (e.g., *Morrison et al., 2008*). $\eta$ represents a fixed learning rate. The synaptic weight is bound such that $w_j \in [0, 1]$. We additionally consider weight-dependent homeostatic mechanisms triggered by pre-

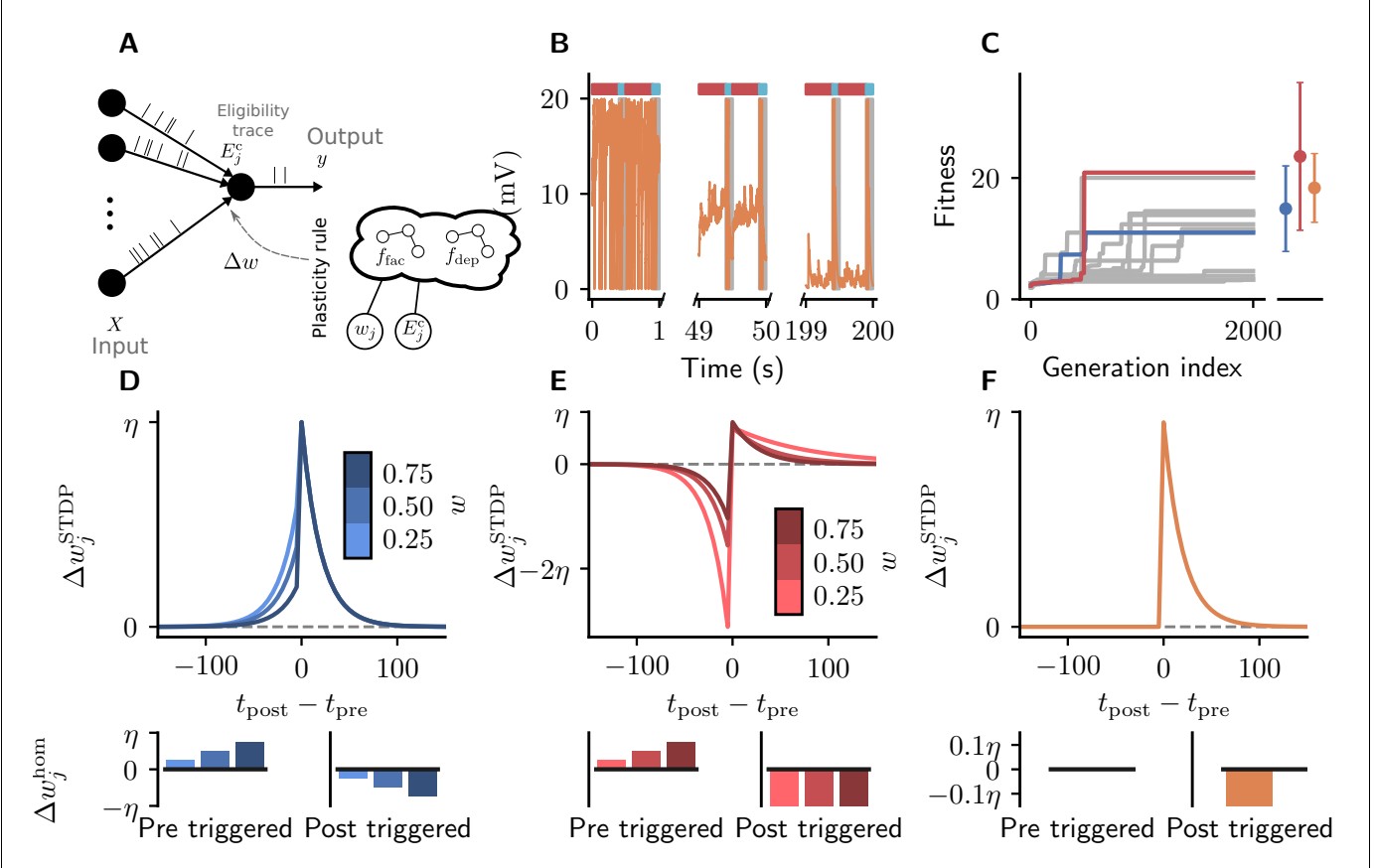

**Figure 5.** Cartesian genetic programming evolves diverse correlation-driven learning rules. (A) Network sketch. Multiple inputs project to a single output neuron. The current synaptic weight $w_j$ and the eligibility trace $E_j^c$ are provided to the plasticity rule that determines the weight update. (B) Membrane potential $u$ of the output neuron over the course of learning using *Equation 17*. Gray boxes indicate presentation of the frozen-noise pattern. (C) Fitness (*Equation 13*) of the best individual per generation as a function of the generation index for multiple runs of the evolutionary algorithm with different initial conditions. Blue and red curves correspond to the two representative plasticity rules selected for detailed analysis. Blue and red markers represent fitness of the two representative rules and the orange marker the fitness of the homeostatic STDP rule (*Equation 17*; *Masquelier, 2018*), respectively, on 20 validation tasks not used during the evolutionary search. Error bars indicate one standard deviation over tasks. (D, E): Learning rules evolved by two runs of CGP (D: LR1, *Equation 19*; E: LR2, *Equation 20*). (F): Homeostatic STDP rule *Equation 17* suggested by *Masquelier, 2018*. Top panels: STDP kernels $\Delta w_j$ as a function of spike timing differences $\Delta t_j$ for three different weights $w_j$. Bottom panels: homeostatic mechanisms for those weights. The colors are specific to the respective learning rules (blue for LR1, red for LR2), with different shades representing the different weights $w_j$. The learning rate is $\eta = 0.01$.

and postsynaptic spikes, respectively. These are implemented by additional functions of the general form:

$$\Delta w_j^{\mathrm{hom}} = \eta \begin{cases} f_{\mathrm{pre}}^{\mathrm{hom}}(w_j) & \text{upon presynaptic spike} \\ f_{\mathrm{post}}^{\mathrm{hom}}(w_j) & \text{upon postsynaptic spike} \end{cases} \tag{16}$$

Weight changes are determined jointly by *Equation 15* and *Equation 16* as $\Delta w_j = \Delta w_j^{\mathrm{STDP}} + \Delta w^{\mathrm{hom}}$. Using CGP, we search for functions $f_{\mathrm{dep}}$, $f_{\mathrm{fac}}$, $f_{\mathrm{pre}}^{\mathrm{hom}}$, and $f_{\mathrm{post}}^{\mathrm{hom}}$ that maximize the fitness $\mathcal{F}(f_{\mathrm{dep}}, f_{\mathrm{fac}})$ (*Equation 13*).

As a baseline we consider a rule described by *Masquelier, 2018* (*Figure 5C*). It is a simple additive spike-timing-dependent plasticity (STDP) rule that replaces the depression branch of traditional

STDP variants with a postsynaptically triggered constant homeostatic term $w^{\text{hom}} < 0$ (**Kempter et al., 1999**). The synaptic weight of the projection from input $j$ changes according to (**Figure 5G**):

$$\Delta w_j^{\text{STDP}} = \eta \begin{cases} 0 & \Delta t_j < 0 \ (\text{anticausal interaction}) \\ E_j^{\text{c}} & \Delta t_j \geq 0 \ (\text{causal interaction}) \, , \end{cases} \tag{17}$$

with homeostatic mechanisms:

$$\Delta w_j^{\text{hom}} = \eta \begin{cases} 0 & \text{upon presynaptic spike} \\ w^{\text{hom}} & \text{upon postsynaptic spike} \, . \end{cases} \tag{18}$$

To illustrate the result of synaptic plasticity following **Equation 17** and **Equation 18**, we consider the evolution of the membrane potential of an output neuron over the course of learning (**Figure 5C**). While the target neuron spikes randomly at the beginning of learning, its membrane potential finally stays subthreshold in between pattern presentations and crosses the threshold reliably upon pattern presentation.

After 2000 generations, half of the runs of the evolutionary algorithm discover high-fitness solutions (**Figure 5D**). These plasticity rules lead to synaptic weight configurations which cause the neuron to reliably detect the frozen-noise pattern. From these well-performing learning rules, we pick two representative examples (**Figure 5D,E**) to analyze in detail. Learning rule 1 (LR1, **Figure 5D**) is defined by the following equations:

$$\Delta w_j^{\text{STDP}} = \eta \begin{cases} -(w_j - 1)E_j^{\text{c}} & \Delta t_j < 0 \\ E_j^{\text{c}} & \Delta t_j \geq 0 \end{cases} , \qquad \Delta w_j^{\text{hom}} = \eta \begin{cases} w_j & \text{upon presyn. spike} \\ -w_j & \text{upon postsyn. spike} \, . \end{cases} \tag{19}$$

Learning rule 2 (LR2, **Figure 5E**) is defined by the following equations:

$$\Delta w_j^{\text{STDP}} = \eta \begin{cases} -E_j^{\text{c}}/w_j & \Delta t_j < 0 \\ (w_j E_j^{\text{c}})^{w_j} & \Delta t_j \geq 0 \end{cases} , \qquad \Delta w_j^{\text{hom}} = \eta \begin{cases} w_j & \text{upon presyn. spike} \\ -1 & \text{upon postsyn. spike} \, . \end{cases} \tag{20}$$

The form of these discovered learning rules and associated homeostatic mechanisms suggests that they use distinct strategies to detect the repeated spatio-temporal pattern. LR1 causes potentiation for small time differences, regardless of whether they are causal or anticausal (note that $-(w_j - 1) \geq 0$ since $w_j \in [0, 1]$). In the Hebbian spirit, this learning rule favors correlation between presynaptic and postsynaptic firing. Additionally, it potentiates synaptic weights upon presynaptic spikes, and depresses them for each postsynaptic spike. In contrast, LR2 implements a similar strategy as the learning rule of **Masquelier, 2018**: it potentiates synapses only for small, positive (causal) time differences. Additionally, however, it pronouncedly punishes anticausal interactions. Similarly to LR1, its homeostatic component potentiates synaptic weights upon presynaptic spikes, and depresses them for each postsynaptic spike.

Note how both rules reproduce important components of experimentally established STDP traces (e.g., **Caporale and Dan, 2008**). Despite their differences both in the form of the STDP kernel as well as the associated homeostatic mechanisms, both rules lead to high fitness, that is, comparable system-level behavior.

Unlike the classical perception of homeostatic mechanisms as merely maintaining an ideal working point of neurons (**Davis and Bezprozvanny, 2001**), in both discovered plasticity rules these components support the computational goal of detecting the repeated pattern. By potentiating large weights more strongly than small weights, the pre-synaptically triggered homeostatic mechanisms support the divergence of synaptic weights into strong weights, related to the repeated pattern, and weak ones, providing background input. This observation suggests that homeostatic mechanisms and STDP work hand in hand to achieve desired functional outcomes, similar to homeostatic terms in stabilized Hebbian rules (**Oja, 1982**; **Miller and MacKay, 1994**). Experimental approaches hence need to take both factors into account and variations in observed STDP curves should be reconsidered from a point of functional equivalence when paired with data on homeostatic changes.

In conclusion, for the correlation-driven task, the evolutionary search discovered a wide variety of plasticity rules with associated homeostatic mechanisms supporting successful task learning, thus enabling new perspectives for learning in biological substrates.

## Discussion

Uncovering the mechanisms of learning via synaptic plasticity is a critical step toward understanding brain (dys)function and building truly intelligent, adaptive machines. We introduce a novel approach to discover biophysically plausible plasticity rules in spiking neuronal networks. Our meta-learning framework uses genetic programming to search for plasticity rules by optimizing a fitness function specific to the respective task family. Our evolving-to-learn approach discovers high-performing solutions for various learning paradigms, reward-driven, error-driven, and correlation-driven learning, yielding new insights into biological learning principles. Moreover, our results from the reward-driven and correlation-driven task families demonstrate that homeostatic terms and their precise interation with plasticity play an important role in shaping network function, highlighting the importance of considering both mechanisms jointly.

The experiments considered here were mainly chosen due to their simplicity and prior knowledge about corresponding plasticity rules that provided us with a high-performance reference for comparison. Additionally, in each experiment, we restricted ourselves to a constrained set of possible inputs to the plasticity rule. Here, we chose quantities which have been previously shown to be linked to synaptic plasticity in various learning paradigms, such as reward, low-pass filtered spike trains, and correlations between pre- and postsynaptic activities. This prior knowledge avoids requiring the evolutionary algorithm to rediscover these quantities but limits the search space, thus potentially excluding other efficient solutions.

A key point of E2L is the compact representation of the plasticity rules. We restrict the complexity of the expressions by three considerations. First, we assume that effective descriptions of weight changes can be found that are not unique to each individual synapse. This is a common assumption in computational neuroscience and based on the observation that nature must have found a parsimonious encoding of brain structure, as not every connection in the brain can be specified in the DNA of the organism (*Zador, 2019*); rather, genes encode general principles by which the neuronal networks and subnetworks are organized and reorganized (*Risi and Stanley, 2010*). Our approach aims at discovering such general principles for synaptic plasticity. Second, physical considerations restrict the information available to the plasticity rule to local quantities, such as pre- and post-synaptic activity traces or specific signals delivered via neuromodulators (e.g., *Cox and Witten, 2019*; *Miconi et al., 2020*). Third, we limit the maximal size of the expressions to keep the resulting learning rules interpretable and avoid overfitting.

We explicitly want to avoid constructing an opaque system that has high task performance but does not allow us to understand how the network structure is shaped over the course of learning. Since we obtain analytically tractable expressions for the plasticity rule, we can analyze them with conventional methods, in contrast to approaches representing plasticity rules with ANNs (e.g., *Risi and Stanley, 2010*; *Orchard and Wang, 2016*; *Bohnstingl et al., 2019*), for which it is challenging to fully understand their macroscopic computation. This analysis generates intuitive understanding, facilitating communication and human-guided generalization from a set of solutions to different network architectures or task domains. In the search for plasticity rules suitable for physical implementations in biological systems, these insights are crucial as the identified plasticity mechanisms can serve as building blocks for learning rules that generalize to the actual challenges faced by biological agents. Rather than merely applying the discovered rules to different learning problems, researchers may use the analytic expressions and prior knowledge to distill general learning principles – such as the computational role of homeostasis emerging from the present work – and combine them in new ways to extrapolate beyond the task families considered in the evolutionary search. Therefore, our evolving-to-learn approach is a new addition to the toolset of the computational neuroscientist in which human intuition is paired with efficient search algorithms. Moreover, simple expressions highlight the key interactions between the local variables giving rise to plasticity, thus providing hints about the underlying biophysical processes and potentially suggesting new experimental approaches.

From a different perspective, while the learning rules found in the experiments described above were all evolved from random expressions, one can also view the presented framework as a hypothesis-testing machine. Starting from a known plasticity rule, our framework would allow researchers to address questions like: assuming the learning rule would additionally have access to variable $x$, could this be incorporated into the weight updates such that learning would improve? The automated procedure makes answering such questions much more efficient than a human-guided manual search. Additionally, the framework is suitable to find robust biophysically plausible approximations for complex learning rules containing quantities that might be non-local, difficult to compute, and/or hard to implement in physical substrates. In particular, multi-objective optimization is suitable to evolve a known, complex rule into simpler versions while maintaining high task performance. Similarly, one could search for modifications of general rules that are purposefully tuned to quickly learn within a specific task family, outperforming more general solutions. In each of these cases, prior knowledge about effective learning algorithms provides a starting point from which the evolutionary search can discover powerful extensions.

The automated search can discover plasticity rules for a given problem that exploit implicit assumptions in the task. It therefore highlights underconstrained searches, be this due to scarcity of biological data, the simplicity of chosen tasks or the omission of critical features in the task design. For instance, without asserting equal average spike rates of background and pattern neurons in the correlation-driven task, one could discover plasticity rules that exploit the rate difference rather than the spatio-temporal structure of the input.

Evolved Plastic Artificial Neural Networks (EPANNs; e.g., *Soltoggio et al., 2018*) and in particular adaptive HyperNEAT (*Risi and Stanley, 2010*), represent an alternative approach to designing plastic neural networks. In contrast to our method, however, these approaches include the network architecture itself into the evolutionary search, alongside synaptic plasticity rules. While this can lead to high-performance solutions due to a synergy between network architecture and plasticity, this interplay has an important drawback, as in general it is difficult to tease apart the contribution of plasticity from that of network structure to high task performance (cf. *Gaier and Ha, 2019*). In addition, the distributed, implicit representation of plasticity rules in HyperNEAT can be difficult to interpret, which hinders a deeper understanding of the learning mechanisms. In machine-learning-oriented applications, this lack of credit assignment is less of an issue. For research into plasticity rules employed by biological systems, however, it presents a significant obstacle.

Future work needs to address a general issue of any optimization method: how can we systematically counter overfitting to reveal general solutions? A simple approach would increase the number of sample tasks during a single fitness evaluation. However, computational costs increase linearly in the number of samples. Another technique penalizes the complexity of the resulting expressions, for example, proportional to the size of the computational graph. Besides avoiding overfitting, such a penalty would automatically remove 'null terms' in the plasticity rules, that is, trivial subexpressions which have no influence on the expressions' output. Since it is a priori unclear how this complexity penalty should be weighted against the original fitness measures, one should consider multi-objective optimization algorithms (e.g., *Deb, 2001*).

Another issue to be addressed in future work is the choice of the learning rate. Currently, this value is not part of the optimization process and all tasks assume a fixed learning rate. The analysis of the reward- and error-driven learning rules revealed that the evolutionary algorithm tried to optimize the learning rate using the variables it had access to, partly generating complex terms that that amount to a variable scaling of the learning rate. The algorithm may benefit from the inclusion of additional constants which it could, for example, use for an unmitigated, permanent scaling of the learning rate. However, the dimensionality of the search space scales exponentially in the number of operators and constants, and the feasibility of such an approach needs to be carefully evaluated. One possibility to mitigate this combinatorial explosion is to combine the evolutionary search with gradient-based optimization methods that can fine-tune constants in the expressions (*Topchy and Punch, 2001*; *Izzo et al., 2017*).

Additionally, future work may involve less preprocessed data as inputs while considering more diverse mathematical operators. In the correlation-driven task, one could for example provide the raw times of pre- and postsynaptic spiking to the graph instead of the exponential of their difference, leaving more freedom for the evolutionary search to discover creative solutions. We expect particularly interesting applications of our framework to involve more complex tasks that are

challenging for contemporary algorithms, such as life-long learning, which needs to tackle the issue of catastrophic forgetting (*French, 1999*) or learning in recurrent spiking neuronal networks. In order to yield insights into information processing in the nervous system, the design of the network architecture should be guided by known anatomical features, while the considered task families should fall within the realm of ecologically relevant problems.

The evolutionary search for plasticity rules requires a large number of simulations, as each candidate solution needs to be evaluated on a sufficiently large number of samples from the task family to encourage generalization (e.g., *Chalmers, 1991*; *Bengio et al., 1992*). Due to silent mutations in CGP, that is, modifications of the genotype that do not alter the phenotype, we use caching methods to significantly reduce computational cost as only new solutions need to be evaluated. However, even employing such methods, the number of required simulations remains large, in the order of $10^3 - 10^4$ per evolutionary run. For the experiments considered here, the computational costs are rather low, requiring $24 - 48$ node hours for a few parallel runs of the evolutionary algorithms, easily within reach of a modern workstation. The total time increases linearly with the duration of a single simulation. When considering more complex tasks which would require larger networks and hence longer simulations, one possibility to limit computational costs would be to evolve scalable plasticity rules in simplified versions of the tasks and architectures. Such rules, quickly evolved, may then be applied to individual instances of the original complex tasks, mimicking the idea of 'evolutionary hurdles' that avoid wasting computational power on low-quality solutions (*So et al., 2019*; *Real et al., 2020*). A proof of concept for such an approach is the delta rule: originally in used in small-scale tasks, it has demonstrated incredible scaling potential in the context of error backpropagation. Similar observations indeed hold for evolved optimizers (*Metz et al., 2020*).

Neuromorphic systems – dedicated hardware specifically designed to emulate neuronal networks – provide an attractive way to speed up the evolutionary search. To serve as suitable substrates for the approach presented here, these systems should be able to emulate spiking neuronal networks in an accelerated fashion with respect to real time and provide on-chip plasticity with a flexible specification of plasticity mechanisms (e.g., *Davies et al., 2018*; *Billaudelle et al., 2019*; *Mayr et al., 2019*).

We view the presented methods as a machinery for generating, testing, and extending hypotheses on learning in spiking neuronal networks driven by problem instances and prior knowledge and constrained by experimental evidence. We believe this approach holds significant promise to accelerate progress toward deep insights into information processing in physical systems, both biological and biologically inspired, with immanent potential for the development of powerful artificial learning machines.

## Materials and methods

### Evolutionary algorithm

We use a $\mu + \lambda$ evolution strategy (*Beyer and Schwefel, 2002*) to evolve a population of individuals towards high fitness. In each generation, λ new offsprings are created from μ parents via tournament selection (e.g., *Miller and Goldberg, 1995*) and subsequent mutation. From these $\mu + \lambda$, individuals the best μ individuals are selected as parents for the next generation (Alg. 4.1). In this work, we use a tournament size of one and a fixed mutation probability $p_{\mathrm{mutate}}$ for each gene in an offspring individual. Since in CGP crossover of individuals can lead to significant disruption of the search process due to major changes in the computational graphs (*Miller, 1999*), we avoid it here. In other words, new offspring are only modified by mutations. We use neutral search (*Miller and Thomson, 2000*), in which an offspring is preferred over a parent with equal fitness, to allow the accumulation of silent mutations that can jointly lead to an increase in fitness. As it is computationally infeasible to exhaustively evaluate an individual on all possible tasks from a task family, we evaluate individuals only on a limited number of sample tasks and aggregate the results into a scalar fitness, either by choosing the minimal result or averaging. We manually select the number of sample tasks to balance computational costs and sampling noise for each task. In each generation, we use the same initial conditions to allow a meaningful comparison of results across generations. If an expression is encountered

that cannot be meaningfully evaluated, such as division by zero, the corresponding individual is assigned a fitness of $-\infty$.

---

**Algorithm 1: Variant of $\mu + \lambda$ evolution strategies used in this study. Note the absence of a crossover step.**

---

**Data:** Initial random parent Population $P_0 = \{p_i\}$ of size μ
$t \leftarrow 0$
**while** $t < n_{\text{generations}}$ **do**
 Create new offspring population $Q_t = \mathrm{CreateOffspringPopulation}(P_t)$
 Combine parent + offspring populations $R_t = P_t \cup Q_t$
 Evaluate fitness of each individual in $R_t$
 Pick $P_{t+1} \subset R_t$ best individuals as new parents
 $t \leftarrow t + 1$
**end**
**Function** CreateOffspringPopulation $(P)$
**begin**
 Offspring population $Q = \{\}$
 **while** $|Q| < \lambda$ **do**
 Choose random subset of $P$ of size $N_{\text{tournament}}$
 Choose best individual in the subset and append to $Q$
 **end**
 **for** $q_i \in Q$ **do**
 Mutate each gene of $q_i$ with mutation probability $p_{\text{mutation}}$
 **end**
 Return $Q$
**end**

---

## HAL-CGP

HAL-CGP (*Schmidt and Jordan, 2020*, https://github.com/Happy-Algorithms-League/hal-cgp, *Jordan, 2021b*) is an extensible pure Python library implementing Cartesian genetic programming to represent, mutate and evaluate populations of individuals encoding symbolic expressions targeting applications with computationally expensive fitness evaluations. It supports the translation from a CGP genotype, a two-dimensional Cartesian graph, into the corresponding phenotype, a computational graph implementing a particular mathematical expression. These computational graphs can be exported as pure Python functions, NumPy-compatible functions (*van der Walt et al., 2011*), SymPy expressions (*Meurer et al., 2017*) or PyTorch modules (*Paszke et al., 2019*). Users define the structure of the two-dimensional graph from which the computational graph is generated. This includes the number of inputs, columns, rows, and outputs, as well as the computational primitives, that is, mathematical operators and constants, that compose the mathematical expressions. Due to the modular design of the library, users can easily implement new operators to be used as primitives. It supports advanced algorithmic features, such as shuffling the genotype of an individual without modifying its phenotype to introduce additional drift over plateus in the search space and hence lead to better exploration (*Goldman and Punch, 2014*). The library implements a $\mu + \lambda$ evolution strategy to evolve individuals (see section Evolutionary algorithm). Users need to specify hyperparameters for the evolutionary algorithm, such as the size of parent and offspring populations and the maximal number of generations. To avoid reevaluating phenotypes that have been previously evaluated, the library provides a mechanism for caching results on disk. Exploiting the wide availability of multi-core architectures, the library can parallelize the evaluation of all individuals in a single generation via separate processes.

## NEST simulator

Spiking neuronal network simulations are based on the 2.16.0 release of the NEST simulator (*Gewaltig and Diesmann, 2007*, https://github.com/nest/nest-simulator; *Eppler, 2021* commit 3c6f0f3). NEST is an open-source simulator for spiking neuronal networks with a focus on large networks with simple neuron models. The computationally intensive propagation of network dynamics is implemented in C++ while the network model can be specified using a Python API (PyNEST; *Eppler et al., 2008*; *Zaytsev and Morrison, 2014*). NEST profits from modern multi-core and multi-node systems by combining local parallelization with OpenMP threads and inter-node communication via the Message Passing Interface (MPI) (*Jordan et al., 2018*). The standard distribution offers a variety of established neuron and plastic synapse models, including variants of spike-timing-

dependent plasticity, reward-modulated plasticity and structural plasticity. New models can be implemented via a domain-specific language (*Plotnikov et al., 2016*) or custom C++ code. For the purpose of this study, we implemented a reward-driven (*Urbanczik and Senn, 2009*) and an error-driven learning rule (*Equation 7*; *Urbanczik and Senn, 2014*), as well as a homeostatic STDP rule (*Equation 17*; *Masquelier, 2018*) via custom C++ code. Due to the specific implementation of spike delivery in NEST, we introduce a constant in the STDP rule that is added at each potentiation call instead of using a separate depression term. To support arbitrary mathematical expressions in the error-driven (*Equation 9*) and correlation-driven synapse models (*Equation 15*), we additionally implemented variants in which the weight update can be specified via SymPy compatible strings (*Meurer et al., 2017*) that are parsed by SymEngine (https://github.com/symengine/symengine; *SymEngine Contributors, 2021*) a C++ library for symbolic computation. All custom synapse models and necessary kernel patches are available as NEST modules in the repository accompanying this study (https://github.com/Happy-Algorithms-League/e2l-cgp-snn (copy archived at swh:1:rev:2f370ba6ec46a46cf959afcc6c1c1051394cd02a), *Jordan, 2021a*).

## Computing systems

Experiments were performed on JUWELS (Jülich Wizard for European Leadership Science), an HPC system at the Jülich Research Centre, Jülich, Germany, with 12 Petaflop peak performance. The system contains 2271 general-purpose compute nodes, each equipped with two Intel Xeon Platinum 8168 processors (2×24 cores) and 12×8 GB main memory. Compute nodes are connected via an EDR-Infiniband fat-tree network and run CentOS 7. Additional experiments were performed on the multicore partition of Piz Daint, an HPC system at the Swiss National Supercomputing Centre, Lugano, Switzerland with 1.731 Petaflops peak performance. The system contains 1813 general-purpose compute nodes, each equipped with two Intel Xeon E5-2695 v4 processors (2×18 cores) and 64 GB main memory. Compute nodes are connected via Cray Aries routing and communications ASIC with Dragonfly network topology and run Cray Linux Environment (CLE). Each experiment employed a single compute node.

## Reward-driven learning task

We consider a reinforcement learning task for spiking neurons inspired by *Urbanczik and Senn, 2009*. Spiking activity of the output neuron is generated by an inhomogeneous Poisson process with instantaneous rate $\phi$ determined by its membrane potential $u$ (*Pfister et al., 2006*; *Urbanczik and Senn, 2009*):

$$\phi(u) := \rho\, e^{\frac{u - u_{\text{th}}}{\Delta u}} . \tag{21}$$

Here, $\rho$ is the firing rate at threshold, $u_{\text{th}}$ the threshold potential, and $\Delta u$ a parameter governing the noise amplitude. In contrast to *Urbanczik and Senn, 2009*, we consider an instantaneous reset of the membrane potential after a spike instead of an hyperpolarization kernel. The output neuron receives spike trains from sources randomly drawn from an input population of size $N$ with randomly initialized weights ($w_{\text{initial}} \sim \mathcal{N}(0, \sigma_w)$). Before each pattern presentation, the output neurons membrane potential and synaptic currents are reset.

The eligibility trace in every synapse is updated in continuous time according to the following differential equation (*Urbanczik and Senn, 2009*; *Frémaux and Gerstner, 2015*):

$$\tau_{\text{M}} \dot{E}_j^{\text{r}} = -E_j^{\text{r}} + \frac{1}{\Delta u}\left[\sum_{s \in y} \delta(t - s) - \phi(u(t))\right]\bar{s}_j(t) , \tag{22}$$

where $\tau_{\text{M}}$ governs the time scale of the eligibility trace and has a similar role as the decay parameter $\gamma$ in policy-gradient methods (*Sutton and Barto, 2018*), $\Delta u$ is a parameter of the postsynaptic cell governing its noise amplitude, $y$ represents the postsynaptic spike train, and $\bar{s}_j(t) = (\kappa * s_j)(t)$ the presynaptic spike train $s_j$ filtered by the synaptic kernel $\kappa$. The learning rate $\eta$ was manually tuned to obtain high performance with the one suggested by *Urbanczik and Senn, 2009*. Expected positive and negative rewards in trial $i$ are separately calculated as moving averages over previous trials (*Vasilaki et al., 2009*):

$$\bar{R}_i^{+/-} = (1 - \frac{1}{m_r})\bar{R}_{i-1}^{+/-} + \frac{1}{m_r}[R_{i-1}]_{+/-} \,, \tag{23}$$

where $m_r$ determines the number of relevant previous trials and $[x]_+ := \max(0,x), [x]_- := \min(0,x)$. Note that $\bar{R}^+ \in [0,1]$ and $\bar{R}^- \in [-1,0]$, since $R \in \{-1,1\}$. We obtain the average reward as a sum of these separate estimates $\bar{R} = \bar{R}^+ + \bar{R}^-; \bar{R} \in [-1,1]$, while the expected absolute reward is determined by their difference $\bar{R}_{abs} = \bar{R}^+ - \bar{R}^-; \bar{R}_{abs} \in [0,1]$.

## Error-driven learning task

We consider an error-driven learning task for spiking neurons inspired by *Urbanczik and Senn, 2014*. $N$ Poisson inputs with constant rates ($r_i \sim \mathcal{U}[r_{min}, r_{max}], i \in [1,N]$) project to a teacher neuron and, with the same connectivity pattern, to a student neuron. As in section Reward-driven learning task, spiking activity of the output neuron is generated by an inhomogeneous Poisson process. In contrast to section Reward-driven learning task, the membrane potential is not reset after spike emission. Fixed synaptic weights from the inputs to the teacher are uniformly sampled from the interval $[w_{min}, w_{max}]$, while weights to the student are all initialized to a fixed value $w_0$. In each trial we randomly shift all teacher weights by a global value $w_{shift}$ to avoid a bias in the error signal that may arise if the teacher membrane potential is initially always larger or always smaller than the student membrane potential. Target potentials are read out from the teacher every $\delta t$ and provided instantaneously to the student. The learning rate η was chosen via grid search on a single example task for high performance with *Equation 7*. Similar to *Urbanczik and Senn, 2014*, we low-pass filter weight updates with an exponential kernel with time constant $\tau_I$ before applying them.

## Correlation-driven learning task

We consider a correlation-driven learning task for spiking neurons similar to *Masquelier, 2018*: a spiking neuron, modeled as a leaky integrate-and-fire neuron with delta-shaped post-synaptic currents, receives stochastic spike trains from $N$ inputs via plastic synapses.

To construct the input spike trains, we first create a frozen-noise pattern by drawing random spikes $\mathcal{S}_i^{pattern} \in [0, T_{pattern}], i \in [0, N-1]$ from a Poisson process with rate ν. Neurons that fire at least once in this pattern are in the following called 'pattern neurons', the remaining are called 'background neurons'. We alternate this frozen-noise pattern with random spike trains of length $T_{inter}$ generated by a Poisson process with rate ν (*Figure 5B*). To balance the average rates of pattern neurons and background neurons, we reduce the spike rate of pattern neurons in between patterns by a factor α. Background neurons have an average rate of $\nu_{inter} = \nu \frac{T_{inter}}{T_{inter}+T_{pattern}}$. We assume that pattern neurons spike only once during the pattern. Thus, they have an average rate of rate of $\nu = \alpha\nu_{inter} + \frac{1}{T_{inter}+T_{pattern}} = \alpha\nu_{inter} + \nu_{pattern}$. Plugging in the previous expression for $\nu_{inter}$ and solving for α yields $\alpha = 1 - \frac{\nu_{pattern}}{\nu_{inter}}$. We choose the same learning rate as *Masquelier, 2018*. Due to the particular implementation of STDP-like rules in NEST (*Morrison et al., 2007*), we do not need to evolve multiple functions describing correlation-induced and homeostatic changes separately, but can evolve only one function for each branch of the STDP window. Terms in these functions which do not vanish for $E_j^c \to 0$ are effectively implementing pre-synaptically triggered (in the acausal branch) and post-synaptically triggered (in the causal branch) homeostatic mechanisms.

## Acknowledgements

We gratefully acknowledge funding from the European Union, under grant agreements 604102, 720270, 785907, 945539 (HBP) and the Manfred Stärk Foundation. We further express our gratitude towards the Gauss Centre for Supercomputing e.V. (https://www.gauss-centre.eu) for co-funding this project by providing computing time through the John von Neumann Institute for Computing (NIC) on the GCS Supercomputer JUWELS at Jülich Supercomputing Centre (JSC). We acknowledge the use of Fenix Infrastructure resources, which are partially funded from the European Union's Horizon 2020 research and innovation programme through the ICEI project under the grant agreement No. 800858. We thank all participants from the HBP SP9 Fürberg meetings for stimulating interactions and Tomoki Fukai for initial discussions and support. We also thank Henrik Mettler and Akos

Kungl for helpful comments on the manuscript. All network simulations carried out with NEST (https://www.nest-simulator.org).

## Additional information

### Funding

| Funder | Grant reference number | Author |
|---|---|---|
| European Commission | 604102 | Jakob Jordan<br>Walter Senn<br>Mihai A Petrovici |
| European Commission | 720270 | Jakob Jordan<br>Walter Senn<br>Mihai A Petrovici |
| European Commission | 785907 | Jakob Jordan<br>Walter Senn<br>Mihai A Petrovici |
| Universität Heidelberg | Manfred Stärk Foundation | Mihai A Petrovici |
| National Centre for Super-computing Applications | | Jakob Jordan<br>Maximilian Schmidt |
| European Commission | 800858 | Jakob Jordan |
| European Commission | 945539 | Jakob Jordan<br>Walter Senn<br>Mihai A Petrovici |

The funders had no role in study design, data collection and interpretation, or the decision to submit the work for publication.

### Author contributions

Jakob Jordan, Maximilian Schmidt, Conceptualization, Resources, Data curation, Software, Formal analysis, Validation, Investigation, Visualization, Methodology, Writing - original draft, Writing - review and editing; Walter Senn, Conceptualization, Resources, Funding acquisition, Project administration, Writing - review and editing; Mihai A Petrovici, Conceptualization, Resources, Formal analysis, Funding acquisition, Investigation, Writing - original draft, Project administration, Writing - review and editing

### Author ORCIDs

Jakob Jordan https://orcid.org/0000-0003-3438-5001
Maximilian Schmidt http://orcid.org/0000-0003-1040-2567
Walter Senn http://orcid.org/0000-0003-3622-0497
Mihai A Petrovici https://orcid.org/0000-0003-2632-0427

### Decision letter and Author response

Decision letter https://doi.org/10.7554/eLife.66273.sa1
Author response https://doi.org/10.7554/eLife.66273.sa2

## Additional files

### Supplementary files

• Transparent reporting form

### Data availability

All data and scripts required to reproduce the manuscript figures, as well as source code, simulation and analysis scripts are publicly available at https://github.com/Happy-Algorithms-League/e2l-cgp-

snn (copy archived at https://archive.softwareheritage.org/swh:1:rev:2f370ba6ec46a46cf959afcc6c1c1051394cd02a).

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

# Appendix 1

## A reward-driven learning
### Full evolution data for different CGP hyperparameter choices

The following tables summarize all evolved plasticity rules for the four different hyperparameter sets used for the reward-driven learning experiments.

**CGP hyperparameter set 0**

| | |
|---|---|
| **Population** | $\mu = 1, p_{\text{mutation}} = 0.035$ |
| Genome | $n_{\text{inputs}} = 3, n_{\text{outputs}} = 1, n_{\text{rows}} = 1, n_{\text{columns}} = 24, l_{\text{max}} = 24$ |
| Primitives | Add, Sub, Mul, Div, Const(1.0), Const(0.5) |
| EA | $\lambda = 4, n_{\text{breeding}} = 4, n_{\text{tournament}} = 1, \text{reorder} = \text{true}$ |
| Other | max generations $= 1000$, minimal fitness $= 500.0$ |

**Discovered plasticity rules for hyperparameter set 0**

| Label | Fitness $\mathcal{F}$ | Expression $f$ |
|---|---|---|
| LR0 | 216.2 | $-E_j^{\text{r}} + E_j^{\text{r}}/R$ |
| LR1 | 73.0 | $(R + E_j^{\text{r}2})/\bar{R}$ |
| LR2 | 216.2 | $E_j^{\text{r}}(R - 1.0)$ |
| LR3 | 221.6 | $E_j^{\text{r}}/(2R + (R + 1.0)(R + \bar{R}) + 1.0)$ |
| LR4 | 234.2 | $-E_j^{\text{r}}(R - 1)(R + \bar{R})$ |
| LR5 | 216.2 | $E_j^{\text{r}}(R - 1)$ |
| LR6 | 69.2 | $4.0E_j^{\text{r}2}/\bar{R} + 2.0E_j^{\text{r}}$ |
| LR7 | 234.2 | $E_j^{\text{r}}(R - 1)(R + \bar{R})/R$ |

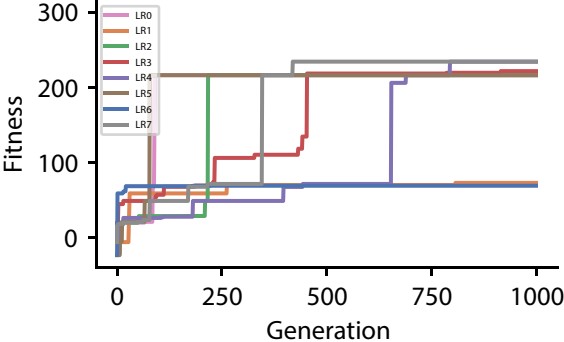

**Appendix 1—figure 1.** Fitness of best individual per generation as a function of the generation index for multiple runs of the evolutionary algorithm with different initial conditions for hyperparameter set 0.

**CGP hyperparameter set 1**

| | |
|---|---|
| **Population** | $\mu = 1, p_{\text{mutation}} = 0.035$ |

*Continued on next page*

*continued*

**CGP hyperparameter set 1**

| Population | $\mu = 1, p_{\text{mutation}} = 0.035$ |
| --- | --- |
| Genome | $n_{\text{inputs}} = \mathbf{4}^*, n_{\text{outputs}} = 1, n_{\text{rows}} = 1, n_{\text{columns}} = \mathbf{12}, l_{\text{max}} = \mathbf{12}$ |
| Primitives | Add, Sub, Mul, Div, Const(1.0), Const(0.5) |
| EA | $\lambda = 4, n_{\text{breeding}} = 4, n_{\text{tournament}} = 1, \text{reorder} = \text{true}$ |
| Other | $\text{max generations} = 1000, \text{minimal fitness} = 500.0$ |

\* Bold highlights values changed with respect to hyperparameter set 0.

**Discovered plasticity rules for hyperparameter set 1**

| Label | Fitness $\mathcal{F}$ | Expression $f$ |
| --- | --- | --- |
| LR0 | 238.6 | $(-E_j^{\text{r}}(R + \bar{R}^-(R + \bar{R}^-)) + E_j^{\text{r}} + \bar{R}^-)/(R + \bar{R}^-(R + \bar{R}^-))$ |
| LR1 | 233.4 | $E_j^{\text{r}}(R - 1)/(R(R - \bar{R}^+))$ |
| LR2 | 217.2 | $-E_j^{\text{r}}(-R + \bar{R}^- + 1.0)$ |
| LR3 | 227.6 | $R\bar{R}^- - E_j^{\text{r}} + E_j^{\text{r}}/R$ |
| LR4 | 247.2 | $(R - 1.0)(R + E_j^{\text{r}} + 2\bar{R}^+)$ |
| LR5 | 198.2 | $(E_j^{\text{r}} - \bar{R}^+ - \bar{R}^-)/(R + \bar{R}^+)$ |
| LR6 | 216.2 | $E_j^{\text{r}}(R - 1)$ |
| LR7 | 225.8 | $-E_j^{\text{r}} - \bar{R}^- + (R - \bar{R}^-)(E_j^{\text{r}} + \bar{R}^-)$ |

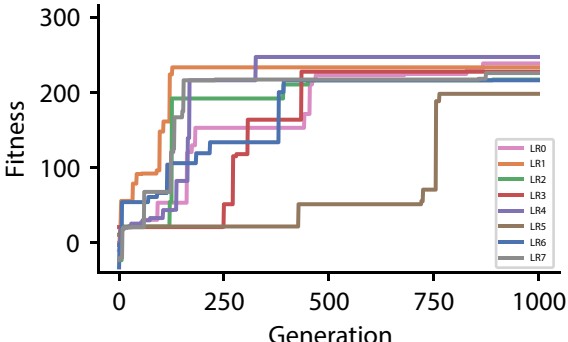

**Appendix 1—figure 2.** Fitness of best individual per generation as a function of the generation index for multiple runs of the evolutionary algorithm with different initial conditions for hyperparameter set 1.

**CGP hyperparameter set 2**

| Population | $\mu = 1, p_{\text{mutation}} = 0.035$ |
| --- | --- |
| Genome | $n_{\text{inputs}} = 4, n_{\text{outputs}} = 1, n_{\text{rows}} = 1, n_{\text{columns}} = \mathbf{24}^*, l_{\text{max}} = \mathbf{24}$ |
| Primitives | Add, Sub, Mul, Div, Const(1.0), Const(0.5) |
| EA | $\lambda = 4, n_{\text{breeding}} = 4, n_{\text{tournament}} = 1, \text{reorder} = \text{false}$ |
| Other | $\text{max generations} = 1000, \text{minimal fitness} = 500.0$ |

\* Bold highlights values changed with respect to hyperparameter set 1.

**Discovered plasticity rules for hyperparameter set 2**

| Label | Fitness $\mathcal{F}$ | Expression $f$ |
|---|---|---|
| LR0 | 127.2 | $E_j^r/(R + \bar{R}^+ - \bar{R}^-)$ |
| LR1 | 192.0 | $E_j^r/(R + \bar{R}^+)$ |
| LR2 | 216.2 | $E_j^r(R - 1)$ |
| LR3 | 170.6 | $(2E_j^r\bar{R}^-(R - \bar{R}^-) + E_j^r - 1)/(R - \bar{R}^-)$ |
| LR4 | 237.6 | $(-RE_j^r(\bar{R}^- + 1) + E_j^r + \bar{R}^-)/(R(\bar{R}^- + 1))$ |
| LR5 | 233.4 | $E_j^r(1 - R)/(R - \bar{R}^+)$ |
| LR6 | 120.8 | $(R + \bar{R}^-)(E_j^r - \bar{R}^+)$ |
| LR7 | 254.8 | $(-R\bar{R}^- + 2E_j^r)(R\bar{R}^- + R - \bar{R}^+)$ |

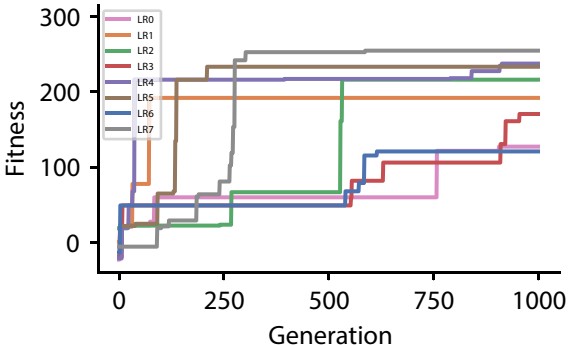

**Appendix 1—figure 3.** Fitness of best individual per generation as a function of the generation index for multiple runs of the evolutionary algorithm with different initial conditions for hyperparameter set 2.

**CGP hyperparameter set 3**

| | |
|---|---|
| **Population** | $\mu = 1, p_{\text{mutation}} = 0.035$ |
| Genome | $n_{\text{inputs}} = 4, n_{\text{outputs}} = 1, n_{\text{rows}} = 1, n_{\text{columns}} = 24, l_{\text{max}} = 24$ |
| Primitives | Add, Sub, Mul, Div, Const(1.0), Const(0.5) |
| EA | $\lambda = 4, n_{\text{breeding}} = 4, n_{\text{tournament}} = 1, \text{reorder} = \text{true}^*$ |
| Other | $\text{max generations} = 1000, \text{minimal fitness} = 500.0$ |

* Bold highlights values changed with respect to hyperparameter set 2.

**Discovered plasticity rules for hyperparameter set 3**

| Label | Fitness $\mathcal{F}$ | Expression $f$ |
|---|---|---|
| LR0 | 236.0 | $E_j^r(-R^3(\bar{R}^- + 1) + 1)/R$ |
| LR1 | 242.0 | $E_j^r(R - \bar{R}^+ + \bar{R}^-)$ |
| LR2 | 242.0 | $E_j^r(R - \bar{R}^+ + \bar{R}^-)$ |

*Continued on next page*

*continued*

**Discovered plasticity rules for hyperparameter set 3**

| Label | Fitness $\mathcal{F}$ | Expression $f$ |
|---|---|---|
| LR3 | 227.6 | $R(E_j^r + \bar{R}^-) - E_j^r$ |
| LR4 | 256.0 | $E_j^r(R - \bar{R}^+ + \bar{R}^-)/(\bar{R}^+ + 1.0)$ |
| LR5 | 71.0 | $(\bar{R}^+(-R + E_j^r + \bar{R}^-(R + \bar{R}^-) + \bar{R}^-) - \bar{R}^-)/\bar{R}^+$ |
| LR6 | 216.2 | $E_j^r(R - 1.0)$ |
| LR7 | 227.8 | $(E_j^r - \bar{R}^{-2})(R + \bar{R}^{-2} - 1.0)$ |

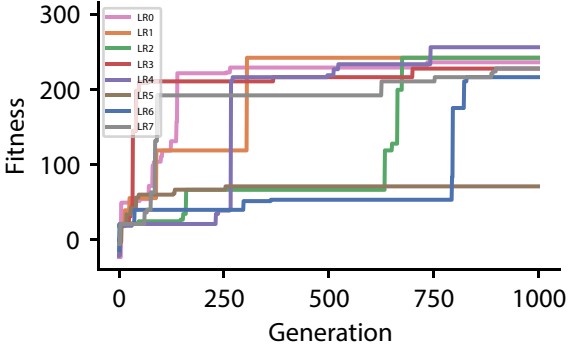

**Appendix 1—figure 4.** Fitness of best individual per generation as a function of the generation index for multiple runs of the evolutionary algorithm with different initial conditions for hyperparameter set 3.

## Causal and homeostatic terms over trials

*Appendix 1—figure 5* illustrates the behavior of the causal and homeostatic terms of the six plasticity rules discovered in the reward-driven learning experiments.

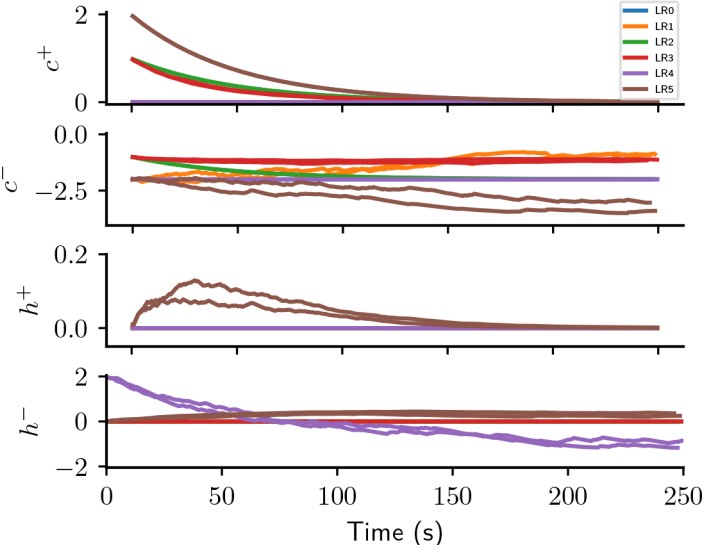

*Appendix 1—figure 5 continued on next page*

**Appendix 1—figure 5.** Causal and homeostatic terms of LR-LR6 over trials. $c^+, c^-$ represent causal terms (prefactors of eligibility trace), $h^+, h^-$ represent homeostatic terms, for positive and negative rewards, respectively.

## Cumulative reward over trials

*Appendix 1—figure 6* illustrates the cumulative reward over trials for the six platicity rules discovered in the reward-driven learning experiments.

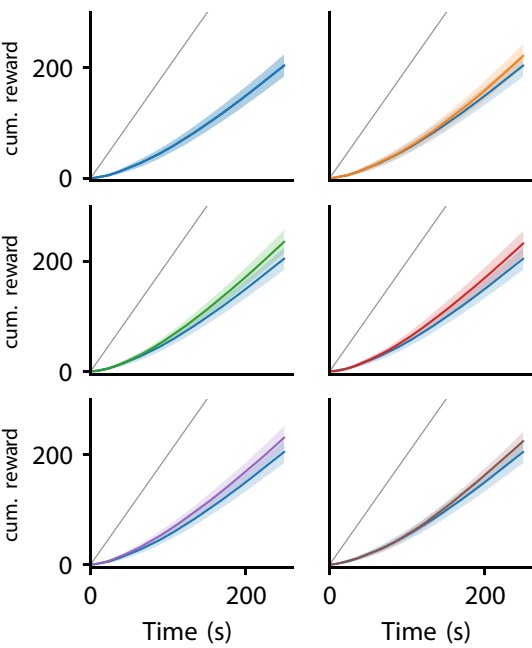

**Appendix 1—figure 6.** Cumulative reward of LR-LR5 over trials. Solid line represent mean, shaded regions indicate plus/minus one standard deviation over 80 experiments. Cumulative reward of LR0 shown in all panels for comparison. Gray line indicates maximal performance (maximal reward received in each trial).

## Error-driven learning – simplification of the discovered rules

As in the main manuscript $v$ is the teacher potential, $u$ the student membrane potential, and $\eta$ a fixed learning rate. $\bar{s}_j(t) = (\kappa * s_j)(t)$ represents the the presynaptic spike train $s_j$ filtered by the synaptic kernel $\kappa$.

We first consider *Equation 10*:

$$\begin{aligned}\Delta w_j &= \eta(v-u)\bar{s}_j\frac{2u-1}{v} \\ &= \eta(v-u)\bar{s}_j\frac{2(v-\delta)-1}{v} \\ &= \eta(v-u)\bar{s}_j\left(2 - 2\underbrace{\frac{\delta}{v}}_{\ll 1} - \underbrace{\frac{1}{v}}_{\approx 0}\right) \\ &\approx 2(v-u)\bar{s}_j\,,\end{aligned}$$

where we introduced $\delta := v - u$. From the third to the fourth line, we assumed that the mismatch between student and teacher potential is much smaller than their absolute magnitude and that their

absolute magnitude is much larger than one. For our parameter choices and initial conditions, this is a reasonable assumption.

We next consider *Equation 11*:

$$
\begin{aligned}
\Delta w_j &= \eta \bar{s}_j (v+u) \frac{v(v-u)-\bar{s}_j}{v^2} \\
&= \eta \bar{s}_j (2v-\delta) \left( \frac{v-u}{v} - \frac{\bar{s}_j}{v^2} \right) \\
&= \eta \bar{s}_j \left( 2 - \frac{\delta}{v} \right) \left( (v-u) - \frac{\bar{s}_j}{v} \right) \\
&= \eta \bar{s}_j \left( \left( 2 - \underbrace{\frac{\delta}{v}}_{\ll 1} \right) (v-u) - 2 \underbrace{\frac{\bar{s}_j}{v}}_{\ll 1} + \underbrace{\frac{\delta}{v}}_{\ll 1} \underbrace{\frac{\bar{s}_j}{v}}_{\ll 1} \right) \\
&\approx 2(v-u)\bar{s}_j
\end{aligned}
$$

As previously, from the third to fourth line, we assumed that the mismatch between student and teacher potential is much smaller than their absolute magnitude and that their absolute magnitude is much larger than one. This implies $\frac{\bar{s}_j}{v} \ll 1$ as $\bar{s}_j \approx \mathcal{O}(1)$ for small input rates.

The additional terms in both *Equation 10* and *Equation 11* hence reduce to a simple scaling of the learning rate and thus perform similarly to the purple rule in *Figure 4*.

## Correlation-driven learning – detailed experimental results

*Appendix 1—figure 7* illustrates membrane potential dynamics for the output neuron using the two plasticity rules discovered in the correlation-driven learning experiments.

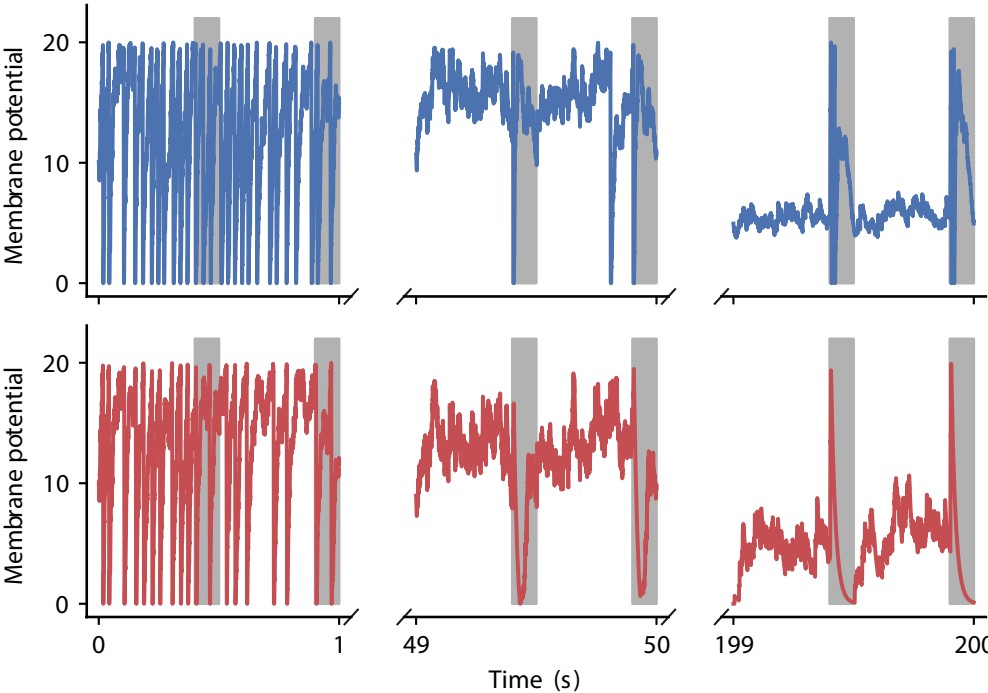

**Appendix 1—figure 7.** Evolution of membrane potential for two evolved learning rules. Membrane potential $u$ of the output neuron over the course of learning using the two evolved learning rules LR1 (top row, *Equation 19*) and LR2 (bottom row, *Equation 20*) (compare *Figure 5B*). Gray boxes indicate presentation of the frozen-noise pattern.

## Simulation details

*Appendix 1—table 1*, *Appendix 1—table 2*, and *Appendix 1—table 3* summarize the network models used in the experiments (according to *Nordlie et al., 2009*).

**Appendix 1—table 1.** Description of the network model used in the reward-driven learning task (4.5).

**A model summary**

| | |
|---|---|
| Populations | 2 |
| Topology | — |
| Connectivity | Feedforward with fixed connection probability |
| Neuron model | Leaky integrate-and-fire (LIF) with exponential post-synaptic currents |
| Plasticity | Reward-driven |
| Measurements | Spikes |

**B populations**

| Name | Elements | Size |
|---|---|---|
| Input | Spike generators with pre-defined spike trains (see 4.5) | $N$ |
| Output | LIF neuron | 1 |

**C connectivity**

| Source | Target | Pattern |
|---|---|---|
| Input | Output | Fixed pairwise connection probability $p$; synaptic delay $d$; random initial weights from $\mathcal{N}(0, \sigma_w^2)$ |

**D neuron model**

| | |
|---|---|
| Type | LIF neuron with exponential post-synaptic currents |
| Subthreshold dynamics | $\frac{\mathrm{d}u(t)}{\mathrm{d}t} = -\frac{u(t) - E_L}{\tau_m} + \frac{I_s(t)}{C_m}$ if not refractory |
| | $u(t) = u_r$ else $I_s(t) = \sum_{i,k} w_k\, e^{-(t - t_i^k)/\tau_s} \Theta(t - t_i^k)$, $k$: neuron index, $i$: spike index |
| Spiking | Stochastic spike generation via inhomogeneous Poisson process with intensity $\phi(u) = \rho\, e^{(u - u_{\mathrm{th}})/\Delta u}$; reset of $u$ to $u_r$ after spike emission and refractory period of $\tau_r$ |

**E synapse model**

| | |
|---|---|
| Plasticity | Reward-driven with episodic update (*Equation 2*, *Equation 3*) |
| Other | Each synapse stores an eligibility trace (*Equation 22*) |

**F simulation parameters**

| | |
|---|---|
| Populations | $N = 50$ |
| Connectivity | $p = 0.8, \sigma_w = 10^3\,\mathrm{pA}$ |
| Neuron model | $\rho = 0.01\,\mathrm{Hz}, \Delta u = 0.2\,\mathrm{mV}, E_L = -70\,\mathrm{mV}, u_r = -70\,\mathrm{mV}, u_{\mathrm{th}} = -55\,\mathrm{mV}, \tau_m = 10\,\mathrm{ms}, C_m = 250\,\mathrm{pF}, \tau_r = 2\,\mathrm{ms}, \tau_s = 2\,\mathrm{ms}$ |
| Synapse model | $\eta = 10, \tau_M = 500\,\mathrm{ms}, d = 1\,\mathrm{ms}$ |
| Input | $M = 30, r = 6\,\mathrm{Hz}, T = 500\,\mathrm{ms}, n_{\mathrm{training}} = 500, n_{\mathrm{exp}} = 10$ |
| Other | $h = 0.01\,\mathrm{ms}, R \in \{-1, 1\}, m_r = 100$ |

**G CGP parameters**

| | |
|---|---|
| Population | $\mu = 1, p_{\mathrm{mutation}} = 0.035$ |
| Genome | $n_{\mathrm{inputs}} = \{3, 4\}, n_{\mathrm{outputs}} = 1, n_{\mathrm{rows}} = 1, n_{\mathrm{columns}} = \{12, 24\}, l_{\mathrm{max}} = \{12, 24\}$ |
| Primitives | Add, Sub, Mul, Div, Const(1.0), Const(0.5) |
| EA | $\lambda = 4, n_{\mathrm{breeding}} = 4, n_{\mathrm{tournament}} = 1, \mathrm{reorder} = \{\mathrm{true, false}\}$ |
| Other | $\mathrm{max\,generations} = 1000, \mathrm{minimal\,fitness} = 500$ |

**Appendix 1—table 2.** Description of the network model used in the error-driven learning task (4.6).

| A model summary | | |
|---|---|---|
| Populations | 3 | |
| Topology | — | |
| Connectivity | Feedforward with all-to-all connections | |
| Neuron model | Leaky integrate-and-fire (LIF) with exponential post-synaptic currents | |
| Plasticity | Error-driven | |
| Measurements | Spikes, membrane potentials | |

| B populations | | |
|---|---|---|
| Name | Elements | Size |
| Input | Spike generators with pre-defined spike trains (see 4.6) | $N$ |
| Teacher | LIF neuron | 1 |
| Student | LIF neuron | 1 |

| C connectivity | | |
|---|---|---|
| Source | Target | Pattern |
| Input | Teacher | All-to-all; synaptic delay $d$; random weights $w \sim \mathcal{U}[w_{\min}, w_{\max}]$; weights randomly shifted by $w_{\text{shift}}$ on each trial |
| Input | Student | All-to-all; synaptic delay $d$; fixed initial weights $w_0$ |

| D neuron model | |
|---|---|
| Type | LIF neuron with exponential post-synaptic currents |
| Subthreshold dynamics | $\frac{\mathrm{d}u(t)}{\mathrm{d}t} = -\frac{u(t)-E_{\mathrm{L}}}{\tau_{\mathrm{m}}} + \frac{I_s(t)}{C_{\mathrm{m}}} I_{\mathrm{s}}(t) = \sum_{i,k} J_k\, e^{-(t-t_i^k)/\tau_{\mathrm{s}}} \Theta(t - t_i^k) k$: neuron index, $i$: spike index |
| Spiking | Stochastic spike generation via inhomogeneous Poisson process with intensity $\phi(u) = \rho\, e^{(u-u_{\mathrm{th}})/\Delta u}$; no reset after spike emission |

| E synapse model | |
|---|---|
| Plasticity | Error-driven with continuous update (**Equation 7**, **Equation 9**) |

| F simulation parameters | |
|---|---|
| Populations | $N = 5$ |
| Connectivity | $w_{\min} = -20, w_{\max} = 20, w_{\text{shift}} \sim \{-15, 15\}, w_0 = 5$ |
| Neuron model | $\rho = 0.2\,\mathrm{Hz}, \Delta u = 1.0\,\mathrm{mV}, E_{\mathrm{L}} = -70\,\mathrm{mV}, u_{\mathrm{th}} = -55\,\mathrm{mV}, \tau_{\mathrm{m}} = 10\,\mathrm{ms}, C_{\mathrm{m}} = 250\,\mathrm{pF}, \tau_{\mathrm{s}} = 2\,\mathrm{ms}$ |
| Synapse model | $\eta = 1.7, d = 1\,\mathrm{ms}, \tau_{\mathrm{I}} = 100.0\,\mathrm{ms}$ |
| Input | $r_{\min} = 150\,\mathrm{Hz}, r_{\max} = 850\,\mathrm{Hz}, T = 10{,}000\,\mathrm{ms}, n_{\exp} = 15$ |
| Other | $h = 0.01\,\mathrm{ms}, \delta t = 5\,\mathrm{ms}$ |

| G CGP parameters | |
|---|---|
| Population | $\mu = 4, p_{\text{mutation}} = 0.045$ |
| Genome | $n_{\text{inputs}} = 3, n_{\text{outputs}} = 1, n_{\text{rows}} = 1, n_{\text{columns}} = 12, l_{\max} = 12$ |
| Primitives | Add, Sub, Mul, Div, Const(1.0) |
| EA | $\lambda = 4, n_{\text{breeding}} = 4, n_{\text{tournament}} = 1$ |
| Other | max generations $= 1000$, minimal fitness $= 0.0$ |

**Appendix 1—table 3.** : Description of the network model used in the correlation-driven learning task (4.7).

**A model summary**

| | |
|---|---|
| Populations | 2 |
| Topology | — |
| Connectivity | Feedforward with fixed connection probability |
| Neuron model | Leaky integrate-and-fire (LIF) with exponential post-synaptic currents |
| Plasticity | Reward-driven |
| Measurements | Spikes |

**B populations**

| Name | Elements | Size |
|---|---|---|
| Input | Spike generators with pre-defined spike trains (see 4.5) | $N$ |
| Output | LIF neuron | 1 |

**C connectivity**

| Source | Target | Pattern |
|---|---|---|
| Input | Output | Fixed pairwise connection probability $p$; synaptic delay $d$; random initial weights from $\mathcal{N}(0, \sigma_w^2)$ |

**D neuron model**

| | |
|---|---|
| Type | LIF neuron with exponential post-synaptic currents |
| Subthreshold dynamics | $\frac{du(t)}{dt} = -\frac{u(t)-E_L}{\tau_m} + \frac{I_s(t)}{C_m}$ if not refractory |

$u(t) = u_r$ else
$I_s(t) = \sum_{i,k} w_k\, e^{-(t-t_i^k)/\tau_s} \Theta(t - t_i^k)$, $k$: neuron index, $i$: spike index

| | |
|---|---|
| Spiking | Stochastic spike generation via inhomogeneous Poisson process with intensity $\phi(u) = \rho\, e^{(u-u_{th})/\Delta u}$; reset of $u$ to $u_r$ after spike emission and refractory period of $\tau_r$ |

**E synapse model**

| | |
|---|---|
| Plasticity | Reward-driven with episodic update (**Equation 2**, **Equation 3**) |
| Other | Each synapse stores an eligibility trace (**Equation 22**) |

**F simulation parameters**

| | |
|---|---|
| Populations | $N = 50$ |
| Connectivity | $p = 0.8, \sigma_w = 10^3\,\text{pA}$ |
| Neuron model | $\rho = 0.01\,\text{Hz}, \Delta u = 0.2\,\text{mV}, E_L = -70\,\text{mV}, u_r = -70\,\text{mV}, u_{th} = -55\,\text{mV}, \tau_m = 10\,\text{ms}, C_m = 250\,\text{pF}, \tau_r = 2\,\text{ms}, \tau_s = 2\,\text{ms}$ |
| Synapse model | $\eta = 10, \tau_M = 500\,\text{ms}, d = 1\,\text{ms}$ |
| Input | $M = 30, r = 6\,\text{Hz}, T = 500\,\text{ms}, n_{training} = 500, n_{exp} = 10$ |
| Other | $h = 0.01\,\text{ms}, R \in \{-1, 1\}, m_r = 100$ |

**G CGP parameters**

| | |
|---|---|
| Population | $\mu = 8, p_{mutation} = 0.05$ |
| Genome | $n_{inputs} = 2, n_{outputs} = 1, n_{rows} = 1, n_{columns} = 5, l_{max} = 5$ |
| Primitives | Add, Sub, Mul, Div, Pow, Const(1.0) |
| EA | $\lambda = 8, n_{breeding} = 8, n_{tournament} = 1$ |
| Other | max generations $= 2000$, minimal fitness $= 10.0$ |

