## [Decision Letter]

**Acceptance summary:**

A precise quantitative description of synaptic plasticity is currently unknown, so that most formulate learning rules somewhat ad hoc. This computational neuroscience paper uses genetic algorithms (GAs) to find synaptic plasticity rules that perform best on a variety of simulated plasticity tasks with spiking neurons. In principle GAs are potentially powerful, but by being non-differentiable they are limited by computational requirements and can only find simple equations and rely on pre-processing such pre-defined eligibility traces. Future research might be able to generalize this technique.

**Decision letter after peer review:**

[Editors’ note: the authors submitted for reconsideration following the decision after peer review. What follows is the decision letter after the first round of review.]

Thank you for submitting your work entitled "Evolving to learn, discovering interpretable plasticity rules for spiking networks" for consideration by *eLife*. Your article has been reviewed by 2 peer reviewers, one of whom is a member of our Board of Reviewing Editors, and the evaluation has been overseen by a Senior Editor. The following individual involved in review of your submission has agreed to reveal their identity: Henning Sprekeler (Reviewer #2).

Our decision has been reached after consultation between the reviewers. Based on these discussions and the individual reviews below, we regret to inform you that your work will not be considered further for publication in *eLife*. Our excuses for the long time we needed to reach our decision.

While the reviewers both saw the potential benefits of the method, the current application only reproduces already known plasticity rules (sometimes not without extra tweaking).

The reviewers agreed that the manuscript does not sufficiently support the applicability of the suggested method to less hand-crafted situations. The scalability of the method is potentially a concern. To be eligible for further consideration in *eLife*, this would have to be shown, either by applying the method to a situation with more and less hand-crafted inputs to the learning rule and/or by identifying novel, efficient and task ensemble-specific rules. Such a revision would likely require more time than *eLife* aims for.

*Reviewer #1:*

This paper uses genetic algorithms to find synaptic plasticity rules.

Overall, I found the results interesting, but as genetic algorithms have been well-established and the amount of new results are limited, I see this more as a tutorial and an effort to bring other researchers to use GAs.

However, the limitations of the method were not clear:

I would like to see a discussion of the computation time, and the scaling with model complexity and other limitations of the technique.

I found it also hard to judge whether this study presents an advance of other methods, such using multiple traces to fit learning rules (Gerstner c.s.).

The inputs to the GAs are quite engineered traces and it was unclear how important this was.

The reworking of the STDP rules in the last Results section was not so clear to me.

First the concept of instability needs to be explained better.

Do these reworked rules perform identically? If so, is there a equivalence class of STDP rules that perform identically on this task?

Furthermore, can stability not be included in the fitness function (either as direct constraint on \Δ w (t->\pm \infty), or by widening the task repertoire)?

It is not so elegant to have a supposedly general technique, and then hand-tune the solutions….

In conclusion, I'm not convinced the study presents enough of a conceptual or methodological advance.

*Reviewer #2:*

The article "Evolving to learn, discovering interpretable learning rules for spiking networks" by Jordan et al. proposes an evolutionary algorithm to meta-learn plasticity rules for spiking neurons. The algorithm learns to combine user-determined inputs into a mathematical formula for updating synaptic weights, by gradually mutating and selecting a set of candidates based on their performance. The algorithm is applied to (families of) reward-based, error-based, and correlation-based tasks, all three performed by a single neuron. In each case, the algorithm recovers previously proposed learning rules (or variants thereof) that are known to optimize some performance measure.

The article is clearly written, timely, and presents an exciting approach to meta-learning that holds the promise of not only generating task-specific learning rules, but of providing them in an interpretable form. Its key weakness is its limitation to a rediscovery of existing general purpose rules, in a setup where the quantities that enter the rules seem somewhat pre-engineered. As such, the paper is a proof-of-concept presentation of a very exciting method on simplistic examples. Whether the approach will actually be applicable to a situation with more inputs, for more complex settings (e.g., multi-layer networks) and whether it will ultimately discover task ensemble-specific learning rules is yet to be seen.

1. Pre-engineering of inputs: It's nice that the authors test their method on three different learning tasks. However, the inputs that enter these learning rules (e.g., the eligibility traces) are chosen by hand and reflect substantial domain knowledge. To show that the method could be applied by a more agnostic user, it would be nice to see that, e.g., different rules could be learned from the same set of inputs. Would it be possible to learn the shape of the eligibility traces? Does the number of successful evolutionary runs decline quickly with the number of inputs the rules are allowed to use?

2. Computational effort: Meta-learning is somewhat notorious in its demand for computing resources, and the authors acknowledge a high-performance computing center. How computationally expensive is the method? How does the computational expense scale with the number of inputs to the learning rules?

3. Discovery of unknown/non-general purpose rules: The paper would be strengthened substantially by an example of a discovered rule that is better than known general purpose rules. I fully appreciate that a search for such a situation may amount to finding a needle in a haystack and I suspect the authors have tried. I nevertheless dare to make a suggestion for a candidate: In Vasilaki et al. (2009), the authors used reward-based spiking learning rules to learn a navigation task and argue that policy-gradient methods fail. What works much better in the end is a biased rule that effectively amounts to something simple like pre x post x reward. Such a rule is clearly biased and could make catastrophic errors in other situations. However, I've suspected for a while (and I think I discussed this with Walter Senn at some point) that such a rule could actually be very powerful in a setting where rewards and state and action representations are very sparse. I wouldn't suggest to the authors to try their method on a navigation task, but policy gradient rules in spiking neurons are notoriously slow in much simpler settings. It feels like it should be possible to beat them. Maybe there is a simple single neuron task with sparse inputs, sparse target outputs and sparse rewards?

4. How sensitive is the method to hyperparameters? The authors use mutation probability 0.045 in first tasks, but 0.05 in last.

[Editors’ note: further revisions were suggested prior to acceptance, as described below.]

Thank you for submitting your article "Evolving interpretable plasticity for spiking networks" for consideration by *eLife*. Your article has been reviewed by 2 peer reviewers, one of whom is a member of our Board of Reviewing Editors, and the evaluation has been overseen by a Reviewing Editor and Michael Frank as the Senior Editor. The following individual involved in review of your submission has agreed to reveal their identity: Henning Sprekeler (Reviewer #2).

Summary:

This computational neuroscience paper uses genetic algorithms (GAs) to find synaptic plasticity rules that perform best on a variety of simulated plasticity tasks with spiking neurons. While GAs are potential powerful, being non-differentiable the study is limited by computationally requirements and can only find very simple equations and use fairly advanced preprocessing such pre-defined eligibility traces.

Essential revisions:

1) The authors have added a number of additions to the manuscript that present clear improvement. However, I have doubts about one of the main additions: the inclusion of different baselines for the RL task. From my understanding, rewards are either 1 or -1. Doesn't that mean that [R]+ = (R+1)/2 and [R]- = (R-1)/2? If that's true, all three baselines are all linearly dependent. My suspicion is that this leads to substantial amounts of "neutral evolution" of terms that basically sum up to zero (or converge to zero exponentially). I juggled around a bit with the various rules and found quite a few of those "null" terms.

I suspect that the point discussed in the section on error-driven learning rules (l. 270) also applies to the RL rules, and that the differences between the rules mostly amount to learning rate (schedules) and "null terms". This may also explain why the gains in performance are not overwhelming. However, because the difference between the evolved rules is not analyzed in depth, this doesn't become clear in the manuscript. I'd suggest to support the discussion of the learning rules by figures. Maybe plot the different "causal" and "homeostatic" terms over time, and potentially something like a running average covariance with (R-1) E?

2) Proper statistical analysis of the findings.

In particular in Figure 3 significance estimates against the null-hypothesis need to be presented.

For that section, I would also like to see if the found learning rules differ in their convergence speed on new data.

3) Include data on the convergence speed of the learning. I still would like to see compute time used for a typical run, which is suspiciously absent from the paper.

*Reviewer #1:*

This paper uses genetic algorithms (GAs) to find synaptic plasticity rules on a variety of plasticity problems with spiking neurons. GAs have as an advantage that a free exploration of possible models potentially coming up with original, superior solutions. On the other hand, being non-differentiable they are severely limited by computationally requirements and can only find very simple equations and use fairly advanced pre-processing such pre-defined eligibility traces.

The paper reads on occasion more like an advertisement than a scientific paper.

For the unsupervised rules, was the LTP made explicitly dependent on (w-1)? Or was this found automatically?

The divergencies described in the previous version of the manuscript seemed to have disappeared like snow in the sun.

*Reviewer #2:*

– The discussion about the need to jointly consider learning rule and homeostatic mechanisms is nice, but of limited novelty. I'd suggest to cite at least Mackay and Miller (1994) here.

– Figure 3: Panel 3 doesn't show any rules with new baselines. Is this old data?

– I failed to find the time window on which the baselines are computed (m=?) This is quite important for your discussion about time varying learning rates.

– Section references are broken. Recompile?

– Double words: l. 82, 377

– l. 364: I learned the hard way to stay away from "first demonstration" claims in papers …

---

## [Author Response]

[Editors’ note: the authors resubmitted a revised version of the paper for consideration. What follows is the authors’ response to the first round of review.]

Reviewer #1:This paper uses genetic algorithms to find synaptic plasticity rules.Overall, I found the results interesting, but as genetic algorithms have been well-established and the amount of new results are limited, I see this more as a tutorial and an effort to bring other researchers to use GAs.

We thank the reviewer for their comments, however, our manuscript is not a tutorial on genetic algorithms. Besides demonstrating for the first time the power of genetic programming in the search for plasticity rules in spiking neuronal networks, it provides new results with implications for both learning algorithm design as well as neuroscientific experiments. We hope that our new results on reinforcement learning and a significant revision of the correlation-driven learning section makes our contributions more evident and improves their accessibility. In the revised version of the manuscript we have addressed all concerns voiced by the reviewer. Please also consider our point-by-point reply below.

However, the limitations of the method were not clear:I would like to see a discussion of the computation time, and the scaling with model complexity and other limitations of the technique.

We agree with the reviewer that the discussion of these topics was rather limited. The Discussion section of the revised manuscript now contains a description of the computational requirements of our methods.

I found it also hard to judge whether this study presents an advance of other methods, such using multiple traces to fit learning rules (Gerstner c.s.).

As we argue in the revised manuscript, our approach presents an advance over previous methods on several fronts. In contrast to traditional approaches relying on mathematical derivations requiring a significant number of assumption besides human intuition and manual work, the automated search requires fewer assumptions, is less biased and mainly consumes machine time. In contrast to previous automated searches for optimization algorithms or plasticity rules relying on representation by artificial neural networks [e.g., 1, 2] we obtain symbolic expressions which allow us to understand the encoded computational principles. In contrast to previous automated searches relying on fixed symbolic expressions with optimized coefficients [e.g., 3, 4] we consider a significantly larger search space by allowing our evolutionary algorithm to manipulate the form of the expressions. But most importantly, the revised manuscript presents new insights into biological information processing, which, we argue, would be difficult to obtain with other methods.

The inputs to the GAs are quite engineered traces and it was unclear how important this was.

The inputs to the plasticity rules are quantities that have been previously been shown to be linked to plasticity, such as reward, low-pass filtered spike trains, and correlations between pre- and postsynaptic activities. The eligibility trace has been successfully used in previous work on reinforcement learning with spiking neuronal networks and is a natural consequence of policy-gradient approaches. We agree with the reviewer that these inputs do reflect domain knowledge and that a possible direction of future research may explore which expressions would arise when one would, for example, provide the individual components of the eligibility trace. However, we are convinced that our algorithm would then be able to recover the current form of the eligibility trace from its components if this form is indeed a prerequisite for high performance.

Moreover, we would like to point out that making use of domain knowledge is potentially desirable and explicitly supported by our approach. We can leverage prior knowledge about the relevance of certain kinds of plasticity-related quantities instead of letting the algorithm “reinvent” them. If the user provides useful variables, the automated search is able to leverage information contained in these to suggest powerful rules. If a certain quantity has been consistently demonstrated to be related to synaptic plasticity in the literature, it seems reasonable to provide it as an input to the plasticity rule if the goal is to achieve high task performance, not the discovery of new “features”, e.g., variants of eligibility traces.

We have expanded the discussion of these points in the revised version of the manuscript.

The reworking of the STDP rules in the last Results section was not so clear to me.First the concept of instability needs to be explained better.Do these reworked rules perform identically? If so, is there a equivalence class of STDP rules that perform identically on this task?

We agree with the reviewer that the presentation of these results was suboptimal. The discovered plasticity rules indeed are functionally equivalent, i.e., despite their difference in mechanism, they lead to similar network-level behavior. The constants in the discovered plasticity rules can indeed be interpreted as pre- and post-synaptically triggered homeostatic mechanisms, suggesting that plasticity and homeostatic principles should always be considered jointly. The revised manuscript contains a more accessible presentation of these results.

Furthermore, can stability not be included in the fitness function (either as direct constraint on \Δ w (t->\pm \infty), or by widening the task repertoire)?

Including such direct constraints in the fitness function would be possible. However, this would require an additional hyperparameter for each constraint which governs how strongly this constraint influences the fitness of an individual. Similar to regularizers in traditional optimization problems, these hyperparameters would require careful tuning which is rather unnatural in our task-focused framework.

Equivalently, a similar effect could be achieved by widening the task repertoire to include specific tasks that effectively implement such constraints. This would, of course, be possible, but would have the same drawback, namely the tuning and weighing of these additional tasks. Another option would be to include multiple natural, but still different tasks, which could implicitly optimize for stability by effectively imposing a higher number of constraints than a single task. This scenario might also require some tuning of the balance between the tasks, but would certainly be more natural and should therefore be explored in future work.

It is not so elegant to have a supposedly general technique, and then hand-tune the solutions….

We thank the reviewer for raising this concern, but we would like to point out that the analysis and potentially even modification of the discovered solutions is an essential part of the workflow enabled by our approach. One main advantage of our method is the availability of compact humanreadable symbolic expressions, in contrast to the plethora of methods relying on artificial neural networks to represent plasticity rules or optimization algorithms. This allows us to analyze the discovered rules with traditional methods to gain an understanding of the computational principles underlying them. We view our approach as a hypothesis-generating machine that suggests possible plasticity rules. By themselves, these could provide new, computationally interesting solutions to learning problems. However, the benefit of our approach does not stop here. Equipped with the knowledge and expertise gained over many decades, researchers in our community can use these automatically generated expressions as starting points for further refinement, recombination and extrapolation. While all plasticity rules in our manuscript were generated by the evolutionary algorithm without hand-tuning, future work may involve manipulating the suggested expressions to incorporate additional expert knowledge not available to the automated search. We have expanded the discussion of these points in the revised manuscript.

In conclusion, I'm not convinced the study presents enough of a conceptual or methodological advance.

We thank the reviewer for their detailed critique of the manuscript and hope that our answers, alongside a substantially revised manuscript including new results, help to better substantiate the relevance of our approach. We hope that the revised manuscript, including new, previously unknown learning rules for reinforcement learning with spiking neurons, make the relevance of our work more accessible.

Reviewer #2:The article "Evolving to learn, discovering interpretable learning rules for spiking networks" by Jordan et al. proposes an evolutionary algorithm to meta-learn plasticity rules for spiking neurons. The algorithm learns to combine user-determined inputs into a mathematical formula for updating synaptic weights, by gradually mutating and selecting a set of candidates based on their performance. The algorithm is applied to (families of) reward-based, error-based, and correlation-based tasks, all three performed by a single neuron. In each case, the algorithm recovers previously proposed learning rules (or variants thereof) that are known to optimize some performance measure.The article is clearly written, timely, and presents an exciting approach to meta-learning that holds the promise of not only generating task-specific learning rules, but of providing them in an interpretable form. Its key weakness is its limitation to a rediscovery of existing general purpose rules, in a setup where the quantities that enter the rules seem somewhat pre-engineered. As such, the paper is a proof-of-concept presentation of a very exciting method on simplistic examples. Whether the approach will actually be applicable to a situation with more inputs, for more complex settings (e.g., multi-layer networks) and whether it will ultimately discover task ensemble-specific learning rules is yet to be seen.

We thank the reviewer for appreciating the novelty of our approach, their valuable comments, and constructive critique. The revised manuscript contains novel results that go beyond merely rediscovering previously know plasticity rules. Below we have addressed the issues raised by the reviewer in a point-to-point reply.

1. Pre-engineering of inputs: It's nice that the authors test their method on three different learning tasks. However, the inputs that enter these learning rules (e.g., the eligibility traces) are chosen by hand and reflect substantial domain knowledge. To show that the method could be applied by a more agnostic user, it would be nice to see that, e.g., different rules could be learned from the same set of inputs. Would it be possible to learn the shape of the eligibility traces? Does the number of successful evolutionary runs decline quickly with the number of inputs the rules are allowed to use?

The inputs to the plasticity rules are quantities that have been previously been shown to be linked to plasticity, such as reward, low-pass filtered spike trains, and correlations between pre- and postsynaptic activities. The eligibility trace has been successfully used in previous work on reinforcement learning with spiking neuronal networks and is a natural consequence of policy-gradient approaches. We agree with the reviewer that these inputs do reflect domain knowledge and that a possible direction of future research may explore which expressions would arise when one would, for example, provide the individual components of the eligibility trace. However, we are convinced that our algorithm would then be able to recover the current form of the eligibility trace from its components if this form is indeed a prerequisite for high performance.

Moreover, we would like to point out that making use of domain knowledge is potentially desirable and explicitly supported by our approach. We can leverage prior knowledge about the relevance of certain kinds of plasticity-related quantities instead of letting the algorithm “reinvent” them. If the user provides useful variables, the automated search is able to leverage information contained in these to suggest powerful rules. If a certain quantity has been consistently demonstrated to be related to synaptic plasticity in the literature, it seems reasonable to provide it as an input to the plasticity rule if the goal is to achieve high task performance, not the discovery of new “features”, e.g., variants of eligibility traces.

The scaling of the evolutionary search with the number of inputs to the expressions heavily depends on a number of factors: the complexity of the “optimal solution”, the types of inputs, e.g., whether they are independent or correlated quantities, the operators available to the search and so on. Naturally, we expect the required runtime to increase with the complexity of the considered problem. We report that so far, we have not encountered insurmountable issues. However, we believe a detailed investigation of how CGP scales along various hyperparameter axes is outside the scope of the current work.

We have expanded the discussion of these points in the revised version of the manuscript and also refer to our answer to the reviewer’s next question.

2. Computational effort: Meta-learning is somewhat notorious in its demand for computing resources, and the authors acknowledge a high-performance computing center. How computationally expensive is the method? How does the computational expense scale with the number of inputs to the learning rules?

The reviewer is correctly pointing out the high demand of computational resources for meta learning. For the experiments considered in the present manuscript, the computational costs are rather low, requiring 24 − 48 node hours for a few parallel runs of the evolutionary algorithms, easily within reach of a modern workstation. However, more complex tasks requiring larger networks and/or longer runs will increase the computational resources and thus require longer waiting times, more parallelism, or faster simulation platforms. In particular, we point to neuromorphic systems as likely candidates for this task. Another option is to evolve rules relying on small networks and simplified task, and only run few instances at full scale for example by considering so called “hurdles” in which a fitness evaluation is stopped early unless the fitness value is not high enough after initial simulations [e.g., 5]. Furthermore, previous work has successfully demonstrated the application of CGP to directly evolve agents for complex tasks, such as playing Atari games, providing some confidence in the method’s scalability [6]. We have expanded the discussion of these points in the revised version of the manuscript.

3. Discovery of unknown/non-general purpose rules: The paper would be strengthened substantially by an example of a discovered rule that is better than known general purpose rules. I fully appreciate that a search for such a situation may amount to finding a needle in a haystack and I suspect the authors have tried. I nevertheless dare to make a suggestion for a candidate: In Vasilaki et al. (2009), the authors used reward-based spiking learning rules to learn a navigation task and argue that policy-gradient methods fail. What works much better in the end is a biased rule that effectively amounts to something simple like pre x post x reward. Such a rule is clearly biased and could make catastrophic errors in other situations. However, I've suspected for a while (and I think I discussed this with Walter Senn at some point) that such a rule could actually be very powerful in a setting where rewards and state and action representations are very sparse. I wouldn't suggest to the authors to try their method on a navigation task, but policy gradient rules in spiking neurons are notoriously slow in much simpler settings. It feels like it should be possible to beat them. Maybe there is a simple single neuron task with sparse inputs, sparse target outputs and sparse rewards?

We agree with the reviewer that discovering new learning principles using the presented approach is what makes it exciting. First, we would like to point out that the initial submission contained such new findings: various forms of STDP kernels lead to similar network-level behavior. However, the presentation of the corresponding results was certainly improvable and we have accordingly completely revised the corresponding section. Second, we thank the reviewer for suggesting to re-examine Vasilaki et al. (2009) [7]. As discussed there and in many previous works [e.g., 8, 9, 10] the choice of a suitable reward baseline is notoriously difficult in policy-gradient learning. We therefore expanded our reinforcement-learning task with a number of new inputs representing different quantities suggested as reward baselines, such as the expected rewards and the expected positive and negative reward, respectively. Exploiting the unbiased search of our evolutionary approach, we were curious to see which learning rules would be discovered. Indeed, the evolutionary search found a variety of interesting plasticity rules for learning from rewards making use of the these new quantities. Our results for the task family considered here can be briefly summarized as follows:

References

– average rewards not distinguishing between positive and negative rewards do not provide a good baseline;

– a quantity loosely related to the “novelty” of a task provides a surprisingly effective adaptive baseline that does not require agents to keep track of expected rewards;

– homeostatic mechanisms and their interplay with activity-regulated plasticity support high performance.

Not only do these discoveries provide insight into computational aspects of reinforcement learning, they also allow specific experimental predictions on the behavioral and neuronal level. To reflect these new results we have completely rewritten the corresponding section in the manuscript. We hope that the revised manuscript makes a more convincing point of the new insights that can be gained via our approach

4. How sensitive is the method to hyperparameters? The authors use mutation probability 0.045 in first tasks, but 0.05 in last.

The method is not specifically sensitive to hyperparameters, and we have not explicitly tuned these. Where possible we have chosen parameters extracted from previous work on CGP [e.g., 11]. However, compared to previous work our genome size was chosen rather small to keep the complexity of the resulting expression low. Complexity penalties introduced in the fitness evaluation could be introduced to make such choices obsolete in the future.

[Editors’ note: what follows is the authors’ response to the second round of review.]

Summary:This computational neuroscience paper uses genetic algorithms (GAs) to find synaptic plasticity rules that perform best on a variety of simulated plasticity tasks with spiking neurons. While GAs are potential powerful, being non-differentiable the study is limited by computationally requirements and can only find very simple equations and use fairly advanced preprocessing such pre-defined eligibilty traces.

We thank the editorial team for preparing this summary. However, we disagree with several points expressed in the summary:

• Non-differential optimization algorithms are not fundamentally limited; the recent years have seen several successful large-scale applications of non-differentiable optimization algorithms [1]. We therefore consider that, while historically factual, many concerns about fundamental limitations of GAs are outdated. Along with an increasing number of researchers, we thus believe that the eld of algorithmic optimization has a lot to gain from the application of Gas in appropriate domains, e.g., non-differentiable search spaces, multidimensional optimization, etc.

• We emphasize throughout the manuscript that it is our explicit goal to discover simple equations: in order to be interpretable, understandable and generalizable by humans; learning rules should not be arbitrarily complex.

• Using \advanced preprocessed quantities" is not a shortcoming of our approach, but an opportunity to leverage prior knowledge, such as eligibility traces which are commonly encountered in reinforcement learning algorithms; if such prior knowledge is intentionally ignored, one is free to supply different (\simpler") signals to the plasticity rule. We would like to point out that this is indeed done in the error-driven learning task.

Essential revisions:1) The authors have added a number of additions to the manuscript that present clear improvement. However, I have doubts about one of the main additions: the inclusion of different baselines for the RL task. From my understanding, rewards are either 1 or -1. Doesn't that mean that [R]+ = (R+1)/2 and [R]- = (R-1)/2? If that's true, all three baselines are all linearly dependent. My suspicion is that this leads to substantial amounts of "neutral evolution" of terms that basically sum up to zero (or converge to zero exponentially). I juggled around a bit with the various rules and found quite a few of those "null" terms.I suspect that the point discussed in the section on error-driven learning rules (l. 270) also applies to the RL rules, and that the differences between the rules mostly amount to learning rate (schedules) and "null terms". This may also explain why the gains in performance are not overwhelming. However, because the difference between the evolved rules is not analyzed in depth, this doesn't become clear in the manuscript. I'd suggest to support the discussion of the learning rules by figures. Maybe plot the different "causal" and "homeostatic" terms over time, and potentially something like a running average covariance with (R-1) E?

Yes, this is the correct definition of positive and negative rewards, these quantities are indeed linearly dependent. In many cases, the new terms do represent an effective scaling of the learning rate, which depends on previously received reward and/or punishment as discussed in the main manuscript (e.g., l176ff, l192ff, l202ff, l215ff). Furthermore, these additional quantities are reflected in terms which can be interpreted as homeostatic mechanisms (e.g., l205ff, l218ff). Results on datasets not seen during evolution (Fig3D, new appendix Fig10) demonstrate that these terms, rather than being “null”, do have an effect on learning performance. We have now included figures in the appendix which visualize the time development of these additional causal and homeostatic terms over time, complementary to the mathematical analysis provided in the main manuscript.

2) Proper statistical analysis of the findings.In particular in Figure 3 significance estimates against the null-hypothesis need to be presented.For that section, I would also like to see if the found learning rules differ in their convergence speed on new data.

We have performed Welch’s T-tests for all learning rules against LR0 using an even larger number of experiments that previously (now 80, previously 30). *p*-values for all discovered learning rules are smaller than floating point precision (*p <* 10^−16^). We have included this information in the main manuscript.

Our defined fitness function implicitly measures convergence speed: the fitness is proportional to the cumulative reward obtained over a fixed number of trials. Naturally, learning rules which converge slower will thus have a lower fitness value. We have now included additional figures in the appendix (Fig11) which compare the convergence speed of the different learning rules on new data.

3) Include data on the convergence speed of the learning. I still would like to see compute time used for a typical run, which is suspiciously absent from the paper.

Regarding convergence speed, please see our answer to the previous question.

The compute time required is explicitly addressed in the discussion (l467ff), quite contrary to being “suspiciously absent”. We believe the required time (24-48h runtime on a single workstation) is within reasonable reach for most labs.

Reviewer #1:This paper uses genetic algorithms (GAs) to find synaptic plasticity rules on a variety of plasticity problems with spiking neurons. GAs have as an advantage that a free exploration of possible models potentially coming up with original, superior solutions. On the other hand, being non-differentiable they are severely limited by computationally requirements and can only find very simple equations and use fairly advanced preprocessing such pre-defined eligibilty traces.The paper reads on occasion more like an advertisement than a scientific paper.

While we agree with the reviewer’s assessment concerning the advantages of GAs, we find it necessary to add some nuance to the asserted drawbacks:

– GAs are not “severely limited by computational requirements” (see answer Summary above).

– Finding simple rules is not a drawback but an explicit goal of our approach (see answer to Summary above).

Furthermore, we believe to have offered scientifically rigorous arguments for all of our claims, based on sound theory and extensive simulations. Should the positive conclusions that we draw from such evidence be subsumed as “advertisement”, then we might disagree on semantics, but wholeheartedly stand by our claims.

For the unsupervised rules, was the LTP made explicitly dependent on (w-1)? Or was this found automatically?The divergencies described in the previous version of the manuscript seemed to have disappeared like snow in the sun.

No, we did not adjust the LTP part manually, these terms were found autonomously by the GA.

The divergences discussed in the previous version of the manuscript have not “disappeared like snow in the sun”. On the contrary: the snow remains completely unchanged, but we do shed additional light on its makeup. More specifically, we reanalyzed the learning rules and in particular their implementation in NEST and discovered a more appropriate interpretation of constant terms (terms not decaying to zero for long intervals between pre- and postsynaptic spiking) namely as pre- and post-synaptically triggered homeostatic mechanism. This is discussed at length in our previous reply letter as well as in our previous manuscript revision.

Reviewer #2:– The discussion about the need to jointly consider learning rule and homeostatic mechanisms is nice, but of limited novelty. I'd suggest to cite at least Mackay and Miller (1994) here.

We thank the reviewer for their critical evaluation and constructive criticism. We have included the reference in the revised manuscript.

– Figure 3: Panel 3 doesn't show any rules with new baselines. Is this old data?

Figure 3C shows data from multiple runs of the GA without access to the new baselines, in that sense it’s similar to the ”old data”.

– I failed to find the time window on which the baselines are computed (m=?) This is quite important for your discussion about time varying learning rates.

The time window is unfortunately only reported in the parameter tables in the appendix (*m*_r_ = 100trials (50s)). In the revised version we mention this number in the main text.

– Section references are broken. Recompile?

We have replaced all broken references, thanks.

– Double words: l. 82, 377

We have fixed the double words, thanks.

– l. 364: I learned the hard way to stay away from "first demonstration" claims in papers …

We have removed the “first demonstration” sentence; we agree this is challenging to assess appropriately.

Reference

1. Such, F. P., Madhavan, V., Conti, E., Lehman, J., Stanley, K. O., & Clune, J. (2017). Deep neuroevolution: Genetic algorithms are a competitive alternative for training deep neural networks for reinforcement learning. arXiv:1712.06567.; Stanley, K. O., Clune, J., Lehman, J., & Miikkulainen, R. (2019). Designing neural networks through neuroevolution. Nature Machine Intelligence, 1(1), 24-35.; Real, E., Aggarwal, A., Huang, Y., & Le, Q. V. (2019, July). Regularized evolution for image classi_er architecture search. In Proceedings of the AAAI conference on arti_cial intelligence (Vol. 33, No. 01, pp. 4780-4789).; Real, E., Liang, C., So, D., & Le, Q. (2020). AutoML-zero: evolving machine learning algorithms from scratch. In International Conference on Machine Learning (pp. 8007-8019). PMLR.